# Magnetic field filtering of the boundary supercurrent in unconventional metal NiTe₂-based Josephson junctions

Tian Le [1,8], Ruihan Zhang [1,2,8], Changcun Li[3], Ruiyang Jiang[1,2], Haohao Sheng[1,2], Linfeng Tu[1,4], Xuewei Cao[4], Zhaozheng Lyu[1,5], Jie Shen [1,6], Guangtong Liu [1,5,6], Fucai Liu [3,7] ✉, Zhijun Wang [1,2] ✉, Li Lu [1,2,5,6] ✉ & Fanming Qu [1,2,5,6] ✉

Topological materials with boundary (surface/edge/hinge) states have attracted tremendous research interest. Additionally, unconventional (obstructed atomic) materials have recently drawn lots of attention owing to their obstructed boundary states. Experimentally, Josephson junctions (JJs) constructed on materials with boundary states produce the peculiar boundary supercurrent, which was utilized as a powerful diagnostic approach. Here, we report the observations of boundary supercurrent in NiTe₂-based JJs. Particularly, applying an in-plane magnetic field along the Josephson current can rapidly suppress the bulk supercurrent and retain the nearly pure boundary supercurrent, namely the magnetic field filtering of supercurrent. Further systematic comparative analysis and theoretical calculations demonstrate the existence of unconventional nature and obstructed hinge states in NiTe₂, which could produce hinge supercurrent that accounts for the observation. Our results reveal the probable hinge states in unconventional metal NiTe₂, and demonstrate in-plane magnetic field as an efficient method to filter out the bulk contributions and thereby to highlight the hinge states hidden in topological/unconventional materials.

Bulk band topology permeates in three- and two-dimensional (3D and 2D) condensed matter, e.g., topological insulators and topological semimetals, and gives rise to gapless surface/edge states. These states result from the well-known bulk-boundary correspondence and are topologically protected[1–3]. In recent years, the concept has been extended to $d$-dimensional higher-order topological systems with ($d$-$m$)-dimensional ($m \geq 2$) metallic hinge or corner states[4–10]. Josephson junctions (JJs)[11] have served as a powerful tool to reveal boundary states in topological materials[12–20] as well as non-topological materials[21–23], where supercurrent distributions modulated by the boundary supercurrent could be discriminated by measuring the interference pattern of the critical supercurrent in a magnetic field.

On the other hand, a new category of unconventional materials (being of obstructed atomic limit) has been proposed with the unconventional nature, where the electrons are located away from the nuclei in crystals[24–31]. As a result of the mismatch between average

¹Beijing National Laboratory for Condensed Matter Physics, Institute of Physics, Chinese Academy of Sciences, Beijing, China. ²School of Physical Sciences, University of Chinese Academy of Sciences, Beijing, China. ³School of Optoelectronic Science and Engineering, University of Electronic Science and Technology of China, Chengdu, China. ⁴School of Physics, Nankai University, Tianjin, China. ⁵Hefei National Laboratory, Hefei, China. ⁶Songshan Lake Materials Laboratory, Dongguan, Guangdong, China. ⁷Yangtze Delta Region Institute (Huzhou), University of Electronic Science and Technology of China, Huzhou, China. ⁸These authors contributed equally: Tian Le, Ruihan Zhang. ✉e-mail: fucailiu@uestc.edu.cn; wzj@iphy.ac.cn; lilu@iphy.ac.cn; fanmingqu@iphy.ac.cn

electronic centers and atomic positions, the obstructed hinge/edge states emerge on the boundary, whose bulk band gaps could be much larger than those of topological materials. In general, both the topological and obstructed boundary states can be used for constructing topological superconductivity and Majorana zero modes with the assistance of superconducting proximity effect (SPE)[31–35]. However, in principle it is a challenge to distinguish the boundary states hidden in a semimetal/metal from bulk states, because both of them are metallic. But for their JJs, the supercurrent on bulk states is expected to suffer larger decoherence and dephasing effects than the boundary states, and therefore, the JJs would exhibit a particular behavior based on the boundary supercurrent channels[12–17].

In this work, we report the observations of the boundary supercurrent in unconventional metal NiTe$_2$-based JJs. Particularly, an in-plane magnetic field (only few tens of millitesla) applied parallel to the Josephson current could filter out the bulk supercurrent and retain the robust boundary supercurrent. Based on a further comparison with a JJ which did not include the hinges of the sample, the effect of an in-plane magnetic field perpendicular/parallel to the Josephson current, and theoretical calculations, these observations could be attributed to obstructed hinge states in the unconventional metal NiTe$_2$. Especially, the magnetic field filtering of the supercurrent functions as a compelling route to acquire the nearly pure boundary supercurrent in topological/unconventional materials-based JJs.

## Results

### NiTe$_2$-based JJ

NiTe$_2$ crystallizes in the CdI$_2$-type trigonal structure with a $P\bar{3}m1$ space group (number 164), as schematically illustrated by the left inset of Fig. 1a. The NiTe$_2$ layers individually stack along the $c$-axis ($C_3$ rotation axis) via van der Waals (vdW) force. It was reported as a type-II Dirac material by ab initio calculations and angle-resolved photoemission spectroscopy measurements[36,37]. Exfoliated NiTe$_2$ nanoplates with a thickness more than 30 nm were used in this work. Figure 1a shows the temperature dependence of the resistance ($R$) for a NiTe$_2$ nanoplate

from room temperature to 1.55 K with a typical metallic behavior. Note that NiTe$_2$ is not superconducting down to 30 mK[38]. Since device D1 shown below was measured around 70 mK, our observations should not be caused by intrinsic superconductivity in NiTe$_2$. The magnetic field dependence of $R$ suggests a nearly non-saturating linear or sublinear magnetoresistance as shown by the right inset of Fig. 1a, which is similar to the bulk materials[36,39].

We fabricated JJs on NiTe$_2$ nanoplates with superconducting electrodes NbTiN as shown in Fig. 1b. JJs of this type have been successfully implemented on many topological materials[14,15,40–45], including the exploration of the boundary states. The current-voltage ($I–V$) characteristics of device D1 is shown in Fig. 1c, indicating a Josephson critical supercurrent ($I_c$) of ~1 μA. When a magnetic field is applied perpendicular to the junction ($B_z$), the superconducting interference pattern (SIP) could be obtained as illustrated in Fig. 2a. The SIP is characterized by the periodic oscillations of $I_c$ as marked by the whitish envelope which separates the superconducting and normal states. We note that the $I_c$ decays very slowly with increasing $|B_z|$, which is in stark contrast to the standard one-slit Fraunhofer-like pattern with the form $|\sin(\pi\Phi/\Phi_0)/(\pi\Phi/\Phi_0)|$ in conventional JJs, denoted by the red line in Fig. 2a, where $\Phi = L_{eff}WB_z$ is the magnetic flux, $L_{eff}$ and $W$ are the effective length and width of the junction, respectively, and $\Phi_0 = h/2e$ is the flux quantum ($h$ is the Planck constant, $e$ is the elementary charge)[11]. A similar phenomenon has been reported on various materials which was attributed to the large boundary (edge or hinge) supercurrent density in the JJs[14–22]. As for JJs, the supercurrent density $J_s$ as a function of position $y$, $J_s(y)$, can be extracted from the $B_z$ dependence of $I_c$, $I_c(B_z)$, through the Fourier transform (Dynes-Fulton approach)[46]. Figure 2b depicts the supercurrent density profile $J_s(y)$ extracted from the $I_c(B_z)$ curves, retrieved from Fig. 2a accordingly (see Supplementary Section I). The center of the junction corresponds to the position $y = 0$. Note that large supercurrent densities appear around $y = \pm 0.7$ μm, which locate at the hinges or side surfaces of the sample and give rise to the boundary supercurrent. Therefore, the SIP on D1 is constituted by the bulk and hinge/side-surface supercurrent.

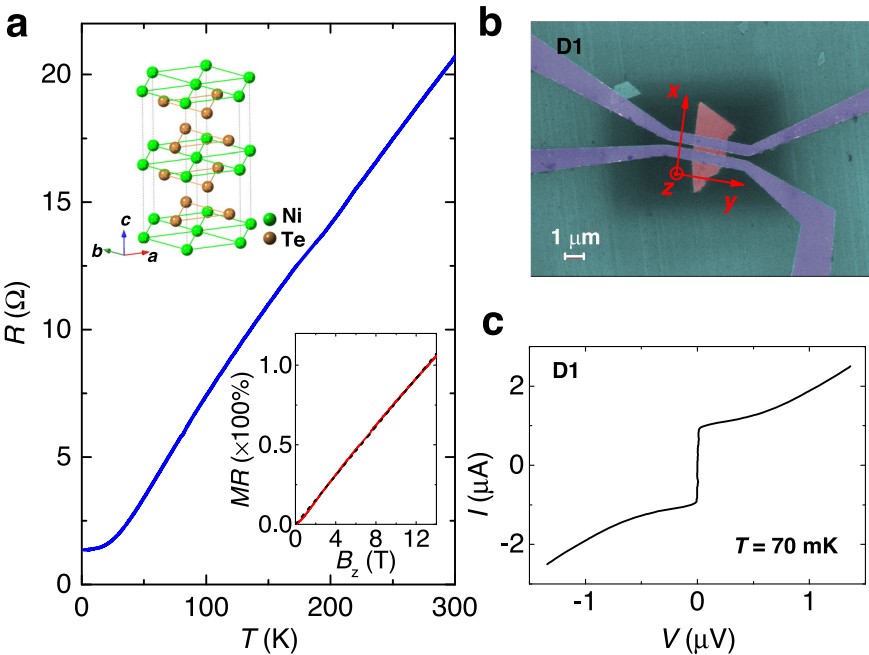

**Fig. 1 | Characterization of the NiTe$_2$-based JJ. a** Temperature dependence of resistance $R$ of an exfoliated NiTe$_2$ nanoplate. The left inset is a schematic illustration of the atomic structure of NiTe$_2$ crystal. The right inset is magnetic field dependence of $R$ at 1.55 K. The black dashed line represents a linear fit to the data. **b** False-color scanning electron microscopic image of a typical NiTe$_2$-based

Josephson junction D1, where the purple color represents superconducting NbTiN electrodes with a width $t$ ~ 500 nm. The separation $L$ between electrodes is ~300 nm. The width $W$ of the NiTe$_2$ nanoplate (red color) between the two electrodes is ~1.5 μm. **c** The $I–V$ characteristic curve showing the Josephson supercurrent at 70 mK.

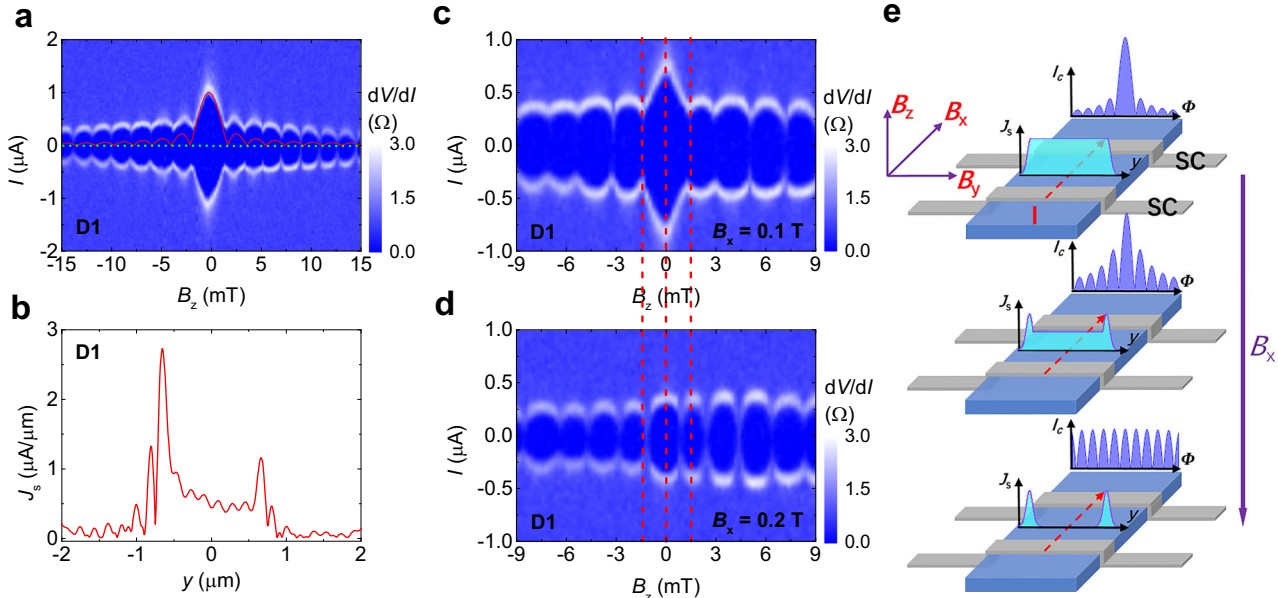

**Fig. 2 | In-plane magnetic field $B_x$ tuning for the SIP. a** SIP for D1 at 70 mK. The red line is a standard Fraunhofer-like curve. **b** Supercurrent density profile $J_s(y)$ for D1 after the Fourier transform of $I_c(B_z)$ in **a**. **c**, **d** SIP for D1 at $B_x = 0.1$ T and 0.2 T, respectively. **e** Schematic illustration of $B_x$-tunable supercurrent density distribution and SIP. The $B_x$ arrow means the increase of magnetic field $B_x$ from top to bottom.

In the following, boundary refers to the hinges or side surfaces of the NiTe$_2$ nanoplate.

### Effect of an in-plane magnetic field

We next investigate the effect of an in-plane magnetic field on the SIP. Figure 2c, d shows the SIPs under $B_x = 0.1$ T and 0.2 T, respectively. Note that the $x$-axis of the 2D color plots has been corrected to eliminate the $B_z$ component induced by $B_{x, y}$ (see Supplementary Section II). Here, $B_x$ denotes the in-plane magnetic field parallel to the Josephson current, while $B_y$ is perpendicular to the Josephson current. The width of the central lobe decreases when increasing $B_x$, as indicated by the red dashed line. Notably, the relative height of the central lobe to the side lobes is strongly suppressed at $B_x = 0.2$ T, indicating a boundary-dominant supercurrent. As sketched in Fig. 2e, when the bulk supercurrent is dominant, it presents a standard Fraunhofer-like pattern with the central lobe possessing a width of $2\Phi_0$ and the side lobe of $\Phi_0$. In particular, the height of the lobes shows a global $1/|B_z|$ fast decay (top row). On the contrary, when the supercurrent flows only along the two hinges/side surfaces (bottom row), the single JJ imitates a superconducting quantum interference device (SQUID) with a two-slit SIP which follows the form $|\cos(\pi\Phi/\Phi_0)|$. In this case, both the central and side lobes have a uniform width of $\Phi_0$ and a weak global decay. If considering the admixture of bulk and boundary supercurrent, the SIP has a finite weight of SQUID signal (middle row), which exactly corresponds to the SIP on our NiTe$_2$-based JJ D1 at $B_x = 0$ T. Accordingly, as shown in Fig. 2d, with the increase of $B_x$ the contribution from the bulk supercurrent decreases significantly and finally a SQUID-like pattern emerges with the boundary-dominant supercurrent. Therefore, the in-plane magnetic field $B_x$ could filter out the bulk supercurrent, i.e., the magnetic field filtering of supercurrent is observed in our experiments.

Such filter effect is crucial for inspecting the contribution from the boundary supercurrent in JJs, even if the weight of the boundary supercurrent is not large enough. Figures 3a, c depict the SIPs for devices D2-1 and D3-1 with different weights of the boundary supercurrent without in-plane magnetic fields. Here, D2-1 displays the deviation from the standard Fraunhofer-like pattern (red line) mainly on the first and second side lobes, while D3-1 shows a very little

deviation. The supercurrent density profiles of Fig. 3a, b shown in the right insets indicate the small weight of the boundary supercurrent. However, an in-plane magnetic field $B_x = 0.04$ T entirely kills the bulk supercurrent and yields SQUID-like patterns as shown in Fig. 3b, c ($B_x$ for killing the bulk supercurrent is device dependent; see Supplementary Section III). In the same way, the supercurrent density profiles in the insets of Fig. 3b, c illustrate the dominance of the boundary supercurrent.

We further compared the effect between $B_x$ and $B_y$ on D4-1 and realized that $B_y$ has a negligible filter effect on the bulk supercurrent. Without the in-plane magnetic field, D4-1 exhibits a SIP with a bulk-dominant supercurrent, as plotted in Fig. 4a. Applying $B_x = 0.04$ T is successful in presenting the SQUID-like pattern as shown in Fig. 4b, as expected. However, the SQUID-like pattern is always absent for $B_y = 0.04$ T, 0.06 T and 0.2 T, as shown in Fig. 4c–e, respectively (The different critical magnetic field of the bulk supercurrent between $B_x$ and $B_y$ is discussed in Supplementary Section IV).

### Origin of the boundary supercurrent

Regarding to the origin of the observed boundary supercurrent, it is commonly attributed to the proximity-induced superconductivity on hinge/side-surface channels[12–23]. However, the bending of the magnetic field lines around the edges of the electrodes was also proposed[47]. In order to further clarify the essential role of the sample hinges/side surfaces, we fabricated a JJ whose junction region did not include the hinges/side surfaces of the sample, as shown by the left inset of Fig. 5a (device D2-2; the upper left junction of D2 shown in the inset of Fig. 3a). The SIP only presents a central lobe as depicted in Fig. 5a, which is merely possible for the Fraunhofer case and could be attributed to the supercurrent extending towards the outside of the junction area due to the superconducting proximity effect. The current density is not uniform as shown by the right inset of Fig. 5a, and thus the SIP is distorted from the standard Fraunhofer pattern and side lobes can be hardly observed. Consistently, the SIP does not show any SQUID-like signal even if applying a magnetic field $B_x = 0.05$ T, as displayed in Fig. 5b (The width of central lobe is smaller than Fig. 5a, primarily due to the large suppression of supercurrent at $B_x = 0.05$ T, which can also be seen in Fig. 4e at $B_y = 0.2$ T). It would be a critical evidence to pin

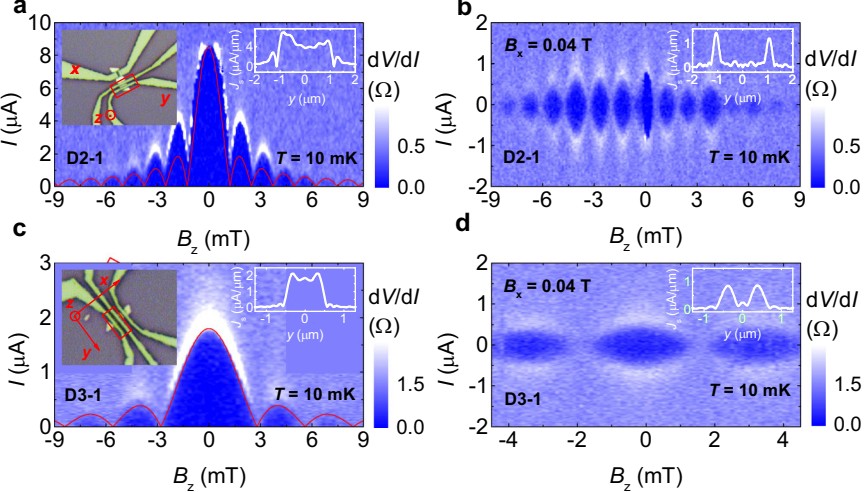

**Fig. 3 | In-plane magnetic field $B_x$ filtered boundary supercurrent on junctions D2-1 and D3-1. a, c** SIP for D2-1 and D3-1 at 10 mK without the in-plane magnetic field, respectively. The red line represents the standard Fraunhofer-like curve. The left insets are optical images for D2-1 and D3-1, indicated by red frames, and with the sample width of 1.9 μm and 1.1 μm, respectively. The right insets depict supercurrent density profiles $J_s(y)$. **b, d** SIP for D2-1 and D3-1 at 10 mK under $B_x = 0.04$ T, respectively. The insets are corresponding supercurrent density profiles $J_s(y)$.

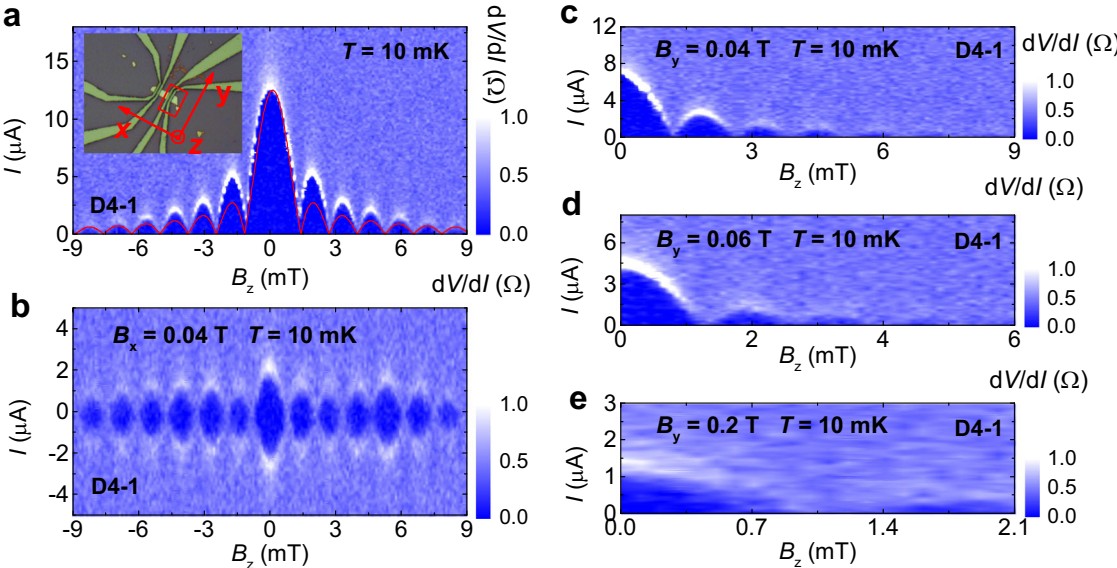

**Fig. 4 | Comparison of the effect of $B_x$ and $B_y$ on SIP for D4-1. a** SIP for D4-1 at 10 mK without the in-plane magnetic field. The red line represents the standard Fraunhofer-like curve. The inset displays the optical image for D4-1, indicated by the red frame, and with the sample width of 2.1 μm. **b** SIP for D4-1 at 10 mK under $B_x = 0.04$ T. **c–e** SIP for D4-1 at 10 mK under $B_y = 0.04$ T, 0.06 T and 0.2 T, respectively.

down the role of the sample hinges/side surfaces for the boundary supercurrent in our JJs.

## The hinge states

A clue of the boundary supercurrent may be found by comparing the assumption of side surfaces and the experimental data (Supplementary Section IV). The measured critical boundary supercurrent at finite $B_y$ is much larger than the calculated values if assuming a side-surface supercurrent that follows the ideal Fraunhofer pattern. Diverse mechanisms could induce deviations, though, it indicates the possibility of hinge supercurrent.

In order to further clarify the origin of boundary supercurrent in NiTe$_2$-based JJs, we next investigate the hinge states in NiTe$_2$ by Density Functional Theory (DFT) calculations. Comparing with the hinge supercurrent originating from the higher-order topology in Cd$_3$As$_2$ and WTe$_2$-based JJs[14–17], it is intriguing to scrutinize the topological

hinge states in NiTe$_2$. However, the type-II Dirac point in NiTe$_2$ is embedded in the bulk bands, and there is no clue yet that it could present topological hinge states.

Instead, our detailed DFT calculations show that NiTe$_2$ has the unconventional nature of charge mismatch, which gives rise to the obstructed hinge states. Meanwhile, the locked spin of the hinge states could explain the observed magnetic field filtering effect of the boundary supercurrent (as shown later). We calculated the NiTe$_2$ rod and obtained the projected spectrum on the hinge atoms shown in the inset of Fig. 6f. To simulate the properties of the NiTe$_2$ monolayer, we slightly enlarged the interlayer distance (only modifying Te-$p_z$ dispersion) and computed the orbital-resolved band structures (Fig. 6a–c) and Wannier charge centers for the lower nine "occupied" bands (Fig. 6d). The results show that the Te-$p_x$, $p_y$ and Ni-$d$ orbitals have a strong hybridization. The Te$^{2-}$ valence state usually means that the Te-$p$ orbitals are fully occupied. Surprisingly, there is a large weight

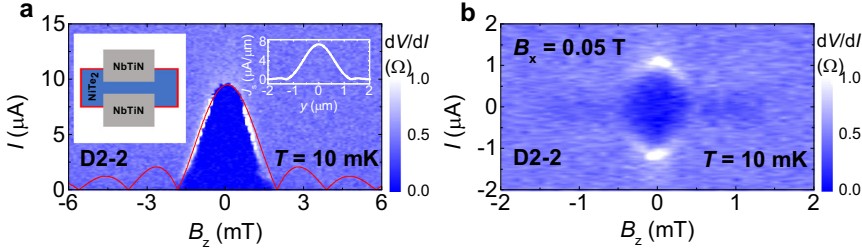

**Fig. 5 | Boundary supercurrent. a** SIP for D2-2 at 10 mK without the in-plane magnetic field. The red line represents the standard Fraunhofer-like curve. The left inset is the schematic illustration of D2-2. The right inset shows the supercurrent density profile $J_s(y)$. **b** SIP for D2-2 at 10 mK under $B_x = 0.05$ T.

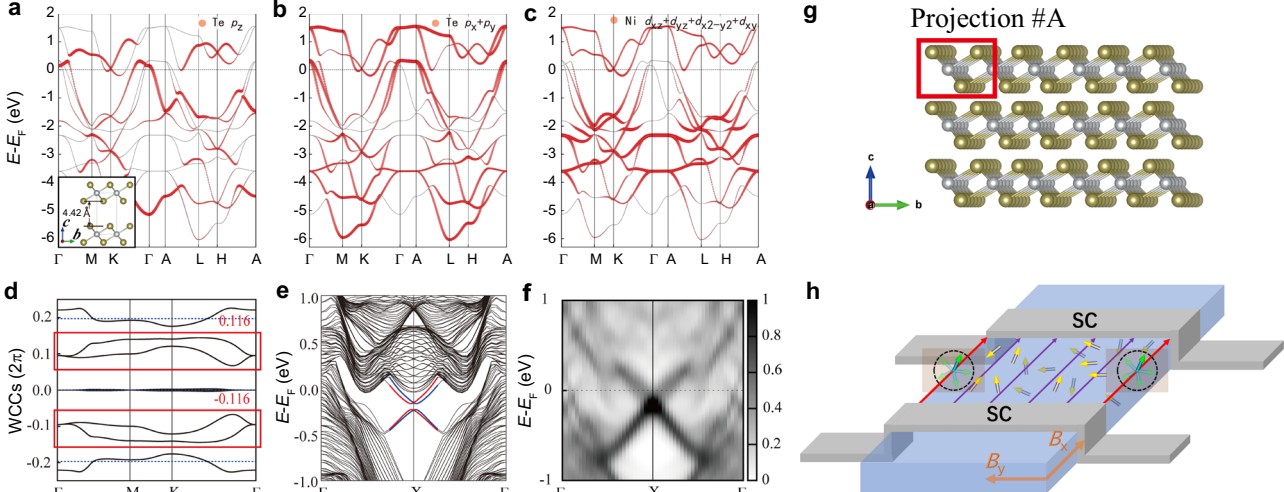

**Fig. 6 | Calculation of the obstructed hinge states. a–c** The orbital-resolved band structure for the modified NiTe$_2$ (with $d_z = 4.42$ Angstrom in the inset), which shows the strong hybridization between the Te-$p_x$, $p_y$ orbitals and Ni-$d$ orbitals. **d** The $z$-directed Wannier charge centers for the occupied nine bands. The dashed lines (0.19c) indicate the locations of the Te atoms. Two Wannier charge centers in the red box are quite away from the Te atoms, with the average of 0.116c. It indicates that the NiTe$_2$ monolayer is unconventional with mismatched electronic charge centers. **e** The obtained obstructed states of the monolayer NiTe$_2$. They are highlighted in the red and blue lines. **f** The hinge spectrum of the pristine NiTe$_2$ ($d_z = 2.63$ Angstrom) with open boundary conditions in both $b$ and $c$ directions. **g** The projected atoms on the hinge. **h** Schematic illustration of the spin distribution. The spin in the bulk is randomized, while on the hinges it is locked to be in the plane perpendicular to the hinges.

of Te-$p_x$/$p_y$ orbitals in the conduction bands, which contradicts with the Te$^{2-}$ valence state. On the other hand, using $z$-directed 1D Wilson loop technique, Wannier charge centers (WCC) are obtained (Fig. 6d), and the two average charge centers in the red box are quite away from the Te atoms (the dashed lines), indicating the unconventional nature of NiTe$_2$ monolayer, which has no symmetry eigenvalue indication[31]. Then, when we performed the calculation in an open boundary condition, the obstructed states were obtained on the edge (Fig. 6e) (red and blue bands indicate the different spin channels due to spin-orbit interactions). To investigate the side surface and hinge states of the bulk NiTe$_2$, we have performed a rod calculation with open boundary conditions in both $b$ and $c$ directions. The results in Fig. 6f show the hinge states of the pristine NiTe$_2$ structure clearly by the projections on the hinge atoms in the red box of Fig. 6g, while the side-surface states are much weaker than the hinge states due to the interlayer hybridization (see Fig. S7 in the Supplementary Section VI). Thus, we demonstrated that the hinge states are regarded as the remnants of the obstructed states of the unconventional metal NiTe$_2$[28,31]. Intriguingly, such unconventional materials were also found to be suitable to construct Josephson diode[27,28,48].

Then, we consider the symmetries on the hinges (in $x$ direction): mirror symmetry ($M_x$) and time reversal ($T$). The $M_x$ makes the spin orientation of hinge states satisfy $\sigma_x(\mathbf{k}) = \sigma_x(-\mathbf{k})$, while $T$ makes $\sigma_x(\mathbf{k}) = -\sigma_x(-\mathbf{k})$. Thus, the combined symmetry $TM_x$ yields $\sigma_x(\mathbf{k}) = 0$

for the non-degenerate hinge states. In other words, due to the coexistence of $M_x$ and $T$, the electron spins of the hinge states are locked to be in the plane perpendicular to the hinges, as illustrated in Fig. 6h. The spin orientation of the non-degenerate hinge states aligns well with the filter effect of the supercurrent under $B_x$, as explained below.

In the case of planar Josephson junctions, when the magnetic field is parallel to the current, the Fraunhofer-like decay of the supercurrent is absent. Considering the small thickness of NiTe$_2$ plates, orbital pair-breaking effect is relatively weak at such small magnetic fields. Therefore, the Zeeman effect should dominate the suppression of the supercurrent. Consequently, the locked spin protects the Cooper pairs of the hinge states from undergoing depairing in $B_x$. In contrary, the spins of the bulk states are randomized without such protection. The coupling between spins and $B_x$ (Zeeman energy) for the bulk is much larger than that for the hinges, and therefore the Cooper pairs of the bulk are easier to break. A similar mechanism has been previously discussed in Ising superconductors to explain the giant upper critical field[49–51]. Due to the spin-orbit-coupling locked spin perpendicular to the MoS$_2$ film, an in-plane upper critical field much larger than the Pauli limit was observed.

Regarding $B_y$, the acquired Zeeman energy to break Cooper pairs for the spins of the hinges is almost the same as that for the bulk. Hence, the Cooper pairs are not protected by the spin-momentum locking. Therefore, the magnetic field filtering of supercurrent is

absent for $B_y$. (Our main train of thought on the hinge states and the filtering effect is shown in Supplementary Section IX.) Recently, modulation of supercurrent induced by the in-plane magnetic field was also observed in graphene systems, which could also be conceptualized as a form of magnetic filtering of supercurrent[52,53].

Next, we discuss the possible mechanisms for the fast suppression of the bulk supercurrent and the anisotropic rate for $B_x$ and $B_y$ (see a more detailed discussion in Supplementary Section III). (1) The fast suppression could be contributed to the Gaussian-like decay of the bulk supercurrent due to orbital spin-flip[54]. However, the anisotropic rate requires an anisotropic diffusion constant ($D$) in NiTe$_2$, which is unknown, suggesting inapplicability. (2) The in-plane magnetic field can lead to Zeeman splitting, resulting in an exponentially suppressed critical current[52]. Again, an anisotropic $D$ is required. (3) A likely mechanism is related to spin-orbital suppression, a manifestation of the interplay between in-plane magnetic field and spin-orbital coupling, and the anisotropy mainly comes from the difference between $B_x$ and $B_y$ on the flux penetration[43]. This mechanism could indeed effectively account for both our observed anisotropic in-plane suppression behavior and the remarkably low critical in-plane field along $B_x$. (4) It might also come from the vimineous shape of the electrodes that exhibit anisotropic demagnetization[55–57]. Note that this mechanism was usually applied to intrinsic superconductors, while in our case it is proximity-induced JJ.

In conclusion, we uncovered the probable hinge supercurrent in NiTe$_2$-based JJs. Our observations combined with the theoretical calculations revealed the unconventional nature and hinge states in NiTe$_2$. In particular, we demonstrated the in-plane magnetic field filtering as a route of vital importance to eliminate unserviceable contributions from bulk states in topological/unconventional materials with hinge states.

## Methods

### Crystal growth and device fabrication

High-quality NiTe$_2$ single crystals were synthesized using the Te flux method. High purity Ni powder (99.9%) and Te ingots (99.99%) with a ratio of 1:10 were sealed in an evacuated quartz tube. Then, the sealed quartz tube was placed in a furnace, and heated to 1000 °C over 10 h. After staying at a constant temperature of 1000 °C for 10 h, the tube was cooled to 600 °C in a rate of 3 °C/h and kept at 600 °C for 3 days to improve the quality of the single crystals. Finally, single crystals of NiTe$_2$ were obtained by removing the remaining Te flux at 600 °C. NiTe$_2$ nanoplates were exfoliated from the single crystals onto a Si/SiO$_2$ wafer in air atmosphere. Ti/NbTiN (5 nm/100 nm) electrodes were deposited via magnetron sputtering with a preceding soft plasma cleaning, using standard electron-beam lithography techniques.

### Transport measurements

Measurements of the temperature and magnetic field dependence of the resistance of NiTe$_2$ nanoplates were carried out with a four-probe configuration in an Oxford TeslatronPT system equipped with a 14 T superconducting magnet. The measurements of Josephson junctions were carried out with a quasi-four-probe configuration in cryofree Oxford Triton dilution refrigerators equipped with a 3-axis vector magnet. A source meter (Keithley 2612B) was used to apply a d.c. bias current $I_{dc}$. A lock-in amplifier (LI5640, NF Corporation) was used to apply a small a.c. excitation current $I_{ac}$ and obtain the differential resistance $dV/dI = V_{ac}/I_{ac}$.

### Theoretical calculation

We performed the first-principles calculations within the framework of the density functional theory (DFT) using projector augmented wave (PAW) method[58,59] implemented in the Vienna ab initio simulation package (VASP)[60,61]. The generalized gradient approximation (GGA) of Perdew-Burke-Ernzerhof (PBE) type[62] was employed for the exchange-

correlation potential. The kinetic energy cutoff for plane wave expansion was set to 400 eV. The thickness of the vacuum layers along $b$ and $c$ directions for the NiTe$_2$ ribbon with open boundary conditions were set to > 20 Å. The Brillouin zone was sampled by Γ-centered Monkhorst-Pack method in the self-consistent process, with a 9 × 9 × 6 $k$-mesh for NiTe$_2$ bulk and a 12 × 1 × 1 $k$-mesh for the NiTe$_2$ ribbon.

## Data availability

The data generated in this study are provided in the Source Data file. Source data are provided in this paper.

## Code availability

The computer code used for DFT calculations is available from Zenodo (https://zenodo.org/records/10694115).

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

## Acknowledgements

We would like to thank Yi Zhou for the fruitful discussions. This work was supported by the National Key Research and Development Program of China (2022YFA1403400, 2022YFA1403800, 2021YFE0194200, 2020YFA0309200, 2017YFA0304700); by the National Natural Science Foundation of China (12074417, 92065203, 92365207, 11974395, 12188101, 12104489, and 11527806); by the Strategic Priority Research Program of Chinese Academy of Sciences (XDB28000000, and XDB33000000); by the Synergetic Extreme Condition User Facility sponsored by the National Development and Reform Commission; and by the Innovation Program for Quantum Science and Technology (2021ZD0302600); and by the Center for Materials Genome.

## Author contributions

F.Q. supervised the project. C.L. synthesized the crystals under the instruction of F.L. T.L., R.J. and L.T. fabricated the devices and performed the transport measurements, with the support from X.C., Z.L., J.S., G.L., L.L. and F.Q. T.L. and F.Q. analysed the data with the help from Z.W. and L.L. R.Z., H.S. and Z.W. did the DFT calculations and symmetry analysis. T.L., F.Q., Z.W., R.Z. and L.L. wrote the manuscript, with input from all authors.

## Competing interests

The authors declare no competing interests.
