## [Peer Review File · Nature Communications]

Magnetic field filtering of the boundary supercurrent in unconventional metal NiTe₂-based Josephson junctionsREVIEWER COMMENTS

Reviewer #1 (Remarks to the Author):

This manuscript “NCOMMS-23-14044-T” entitled “Magnetic field filtering of the hinge supercurrent in unconventional metal NiTe₂-based Josephson junctions” studied the side surfaces/hinge state in a metal NiTe₂. By fabricating NbTiN-NiTe₂-NbTiN Josephson junctions (JJs) and analyzing the non-standard supercurrent interference patterns (SIPs), the authors found the coexistence of bulk and side surfaces/hinge supercurrents, and furthermore the bulk supercurrent can be suppressed by the magnetic field along specific directions while side surfaces/hinge supercurrent remains almost the same. Such topic and novelty fit in the scope of Nature Communications, and I would like to recommend publishing if the authors can address my concerns below:

1. The R-T curve of an exfoliated NiTe₂ nanoplate down to 1.55K showed its metallic nature, yet all the JJs were measured down to 10mK. I am wondering if the NiTe₂ itself is superconducting or not, could the authors cite reference or measure R-T by themselves to reveal its state at 10mK?
2. In fig 2, where the authors claimed the B_x can suppress the bulk supercurrent, my concern is the B_x is too large, even a tiny misalignment between the sample and B_x plane will lead to a considerable large magnetic field at z direction. For example, if there exists 5 degrees misalignment, 0.2T B_x magnetic field will give rise to a ~17mT B_z constant magnetic field. Therefore, the +-9mT B_z region shown in fig 2d is equivalent to the 12-26mT region of fig 2a plus the influence of the remain real x direction magnetic field. And hence, due to the absence of the central peak, one can always obtain the edge dominated current profile by the method in supplementary section I.
3. The SIP in a JJ is largely dependent on the direction of the magnetic field and the shape of the sample, as discussed/observed by many previous works (Nat Nanotechnol 17, 39-44 (2022), J. Appl. Phys. 103, 07C707 (2008), Phys. Rev. B 107 094522 (2023)). Both the heights and periods of the lobes will be influenced by the magnetic field along another direction. Therefore, by applying a constant magnetic field along one direction, and measuring SIP with magnetic field along another direction is not suitable to extract the current profile by the method described in supplementary section I.
4. The authors mentioned that the I_c should show a 1/|B_z| global decay in page 6 and supplementary IV, and extracted the boundary I_c for different lobes, and hence claimed the observation of the hinge state in NiTe₂. The authors should explain it in more detail, and cite references. I cannot fully judge anything here without further explanation and detailed equations.
5. In line 157, the authors claimed the bulk supercurrent behavior with Fig. 5a and 5b. But with only one lobe shown in the SIPs, one can hardly say it's from the Fraunhofer pattern or SQUID pattern. Better SIPs should be provided to support the claim.
6. I noticed a B_x arrow at the rightmost position in Fig. 2, what does that mean? And there is a white square in Fig. 6b which covers some band structures.

Reviewer #2 (Remarks to the Author):

This experimental work discusses the measurement of Type-II Dirac semimetal NiTe₂-based Josephson junction. Authors investigated magnetic field interference pattern of Josephson junctions and suggest that Type-II Dirac semimetal NiTe₂ have hinge states, which mediate supercurrent. When a perpendicular magnetic field is applied to the Josephson junction, the magnetic field interference pattern (Fraunhofer pattern) shows slowly vanishing behavior, suggesting an edge-enhanced current. When an in-plane magnetic field is added, the Fraunhofer pattern is transformed into that of a superconducting quantum interference device (SQUID). Analysis of the magnetic field interference pattern and its Josephson current density shows an unconventional interference pattern and claims the existence of hinge states in NiTe₂.

I have a few questions:

1. Authors mentioned “the magnetic field filtering of the supercurrent functions as a compelling route to acquire the nearly pure boundary supercurrent in topological/unconventional materials based JJs.” in lines 64-67. Could authors explain why and how that “magnetic field filtering” happens in detail? Is there any reference for this mechanism?
2. Authors mentioned, “the magnetic field filtering of supercurrent is absent for B_y .” Could authors explain why magnetic field filtering for B_y is ineffective and how this relates to the magnetic field filtering effect?
3. There are at least three relevant works, including [Nature Phys 13, 87–93 (2017)], [Nat Commun 9, 3478 (2018)], and [Nat. Phys. 18, 1228–1233 (2022)]. Especially, [Nat. Phys. 18, 1228–1233 (2022)] have already investigated the magnetic field interference with an in-plane magnetic field (B_x) parallel to the Josephson current and with the perpendicular magnetic field (B_z), like the work by authors. I wonder how authors’ works are significantly different from those previous works.

Reviewer #3 (Remarks to the Author):

The manuscript by Le and co-authors reports on measurements of Josephson devices consisting of NbTiN superconducting contacts and exfoliated NiTe₂ flakes as weak links. The authors use standard out-of-plane interference patterns to map the current distribution within the junction. The main result of this work is found upon application of very small magnetic fields directed in the plane of the sample. Fields applied parallel to the current flow, in the range of < 0.1 T, cause the supercurrent to re-arrange such that the zero-field Fraunhofer pattern disappears, giving rise to a SQUID-like oscillation. The latter indicates that current is being carried along the edges – an effect which the authors associate with the presence of boundary states. In this sense, the in-plane magnetic field is being used as a way to filter between distinct carrier channels – with the edge channel being immune to its effects.

The authors argue that the origin of the boundary supercurrent is in fact due to hinges, rather than side edges, by presenting the dependence on the field B_y , oriented perpendicular to the current propagation direction. They finally present a set of calculations, suggesting the existence of hinge-modes of obstructed-Wannier orbital type.

This manuscript is very well written, it is clear, and I find it quite convincing. If indeed the authors observe hinge modes, this would be a nice advance over existing state of the art. If the authors can answer the following points, this work can be considered for publication in Nature Communications.

1. My main concern is related to the proof of transport being carried by hinge modes. There are, by now, several works which report supercurrent transport at edges, even such which are not topological, and the transition from bulk to edge due to applied in-plane field is by no means a proof the presence of special edge modes. Edge current can be related to edge accumulation or other effects – e.g. in graphene (Allen et al., Ref 20, and Zhu et al., Nature Comm. 8 14552 (2017), which should be cited here), and also in Bi₂O₂Se (Ref 21).

2. A field-induced transition of supercurrent to a narrow channel (at the edge or not) may also seen in graphene: See e.g. PRB 103, 115401 (2021), and in arxiv: 2211.01020. There works too, I believe, should be cited here.

3. There should be 2 hinges on each side – giving rise to a beating effect. Is there any indication for such beating?

4. The reason why bulk states disappear at such low in-plane field is not properly explained. What drives the difference between individual samples? I don't really understand why the fabrication process should have any role, as suggested in the Supplementary.

Minor Comments:

5. On page 6, better clarify from the 1D hinges, rather than from the 2D vertical side-surfaces.

6. Line 32 – “futile” bulk contributions. I am not sure the work “futile” is the best choice here.

7. Line 37 – “which is a well-known bulk-boundary correspondence” – this entire first sentence does not make grammatical sense.

8. Line 40 – “Josephson junctions have served as a powerful tool to reveal boundary states in topological materials” – citations 11-21. The authors should note that some of their citations are graphene systems, which are not topological. The text should be clear on this.

9. Line 211 – the word “remanence” – I assume the authors intended to use “remnants”, or “a remnant of”.

10. In Line 217, I didn't understand what is the orientation of the locked spins – a diagram should be added.

11. The term “Obstructed hinge states” is used early on – I think this term should be defined in the text in the early part of the manuscript.

Reviewer #4 (Remarks to the Author):

In this work, the authors fabricated a NiTe-based Josephson junction by coupling a NiTe thin film to

two superconducting NbTiN electrodes. The Fraunhofer patterns of the Josephson junctions (critical current as a function of out-of-plane magnetic field) were observed under different conditions. The authors observed that:

1. Edge transport occurred even in the absence of an in-plane magnetic field, as shown in Fig. 2a and the current distribution in Fig. 2b.
2. The bulk supercurrent transport is suppressed by a magnetic field along the current direction, and the edge transport becomes more prominent, as shown in Figs. 2c and 2d.
3. From Figs. 3c and 3d, a small magnetic field of 0.04T is enough to suppress the bulk supercurrent, and the edge supercurrent can dominate the transport.
4. On the other hand, the effect of B_y is very different from that of B_x . A small B_y is not enough to suppress the bulk supercurrent, as shown in Figs. 4c to 4e.
5. The authors showed by DFT calculations that there are hinge states in NiTe.

The experimental results of this work are quite nice. However, I would like the authors to explain a few points more clearly before I can make a recommendation.

1. NiTe is metallic and there are supposed to be a large number of bulk conducting channels. Why can the hinge states dominate the transport as shown in Fig. 2b? From Fig. 2b, it is clear that there are significant edge contributions to the supercurrent.
2. Why can a small magnetic field along the supercurrent suppress the bulk supercurrent so much? This is not explained well in the manuscript. Instead of some handwaving arguments, can the authors provide a more rigorous theoretical support? The current level of reasoning is certainly not enough for Nature Communications. I am not aware of similar effects of magnetic fields suppressing bulk supercurrents. Have similar effects been observed in other Josephson junctions?
3. Similarly, why is there such a large difference between the effects of B_y and B_x ? Can the authors provide a more rigorous theory?
4. From the Fraunhofer pattern and the current distribution, the authors can probably claim that there are edge states in NiTe. However, it is hard to identify them as hinge states. How can the authors rule out the possibility of having edge states? What is the localization length of the hinge states on the side surfaces? How can the authors distinguish the hinge states from edge states for such thin samples?

Overall, the experimental results are quite good. However, the theoretical explanation is not clear enough. Currently, there is not enough experimental nor theoretical evidence to support the presence of hinge states.

We thank all four reviewers for their precious time and valuable comments. The very constructive comments and suggestions helped us a lot to improve our manuscript. The followings are our point-by-point responses to the reviewers' reports. The words in **black** font color are the reviewers' reports, in **blue** font are our responses. The modifications in the manuscript and the Supplementary Information are indicated in **red**.

We found that there are several common questions from different reviewers. So, before replying the reviewers' reports point by point, we would like to summarize these common questions here, i.e., **the proof of hinge states** (Reviewer #1 Comment 4, Reviewer #3 Comment 1, Reviewer #4 Comment 4-1), **detailed mechanism of the magnetic field filtering effect** (Reviewer #2 Comments 1&2, Reviewer #4 Comment 3), and **the low critical field to suppress the bulk supercurrent** (Reviewer #3 Comment 4-1, Reviewer #4 Comment 2-1). We sincerely appreciate all reviewers for raising these crucial questions, which helped us a lot to improve the manuscript and to reach a better presentation. **Modifications #3 (starting from paragraph 2, page 7) and #7 (starting from paragraph 2, page 9) in the revised main text and #10 in the Supplementary Information Section III** are our main responses to these comments. And for a more detailed explanation please refer to the responses to the specific comments below. Please note that for ease of reading, part of our responses will be repeated when answering similar questions.

Reviewer #1

This manuscript "NCOMMS-23-14044-T" entitled "Magnetic field filtering of the hinge supercurrent in unconventional metal NiTe₂-based Josephson junctions" studied the side surfaces/hinge state in a metal NiTe₂. By fabricating NbTiN-NiTe₂-NbTiN Josephson junctions (JJs) and analyzing the non-standard supercurrent interference

patterns (SIPs), the authors found the coexistence of bulk and side surfaces/hinge supercurrents, and furthermore the bulk supercurrent can be suppressed by the magnetic field along specific directions while side surfaces/hinge supercurrent remains almost the same.

Reply: We sincerely thank Reviewer #1 for reading our manuscript so carefully and giving valuable comments and suggestions on our work, which helped us a lot to improve the manuscript. Below please find our detailed responses.

Such topic and novelty fit in the scope of Nature Communications, and I would like to recommend publishing if the authors can address my concerns below.

Reply: We thank Reviewer #1 for these positive comments and for the recommendation of publication.

1. The R-T curve of an exfoliated NiTe₂ nanoplate down to 1.55 K showed its metallic nature, yet all the JJs were measured down to 10 mK. I am wondering if the NiTe₂ itself is superconducting or not, could the authors cite reference or measure R-T by themselves to reveal its state at 10 mK?

Reply: We thank Reviewer #1 for raising this issue. We found that NiTe₂ is not superconducting down to 30 mK, which has been reported by Varnava D. Esin *et al* [Nanomaterials 12, 4114 (2022)]. Since our device D1 was measured around 70 mK, our observations should not be caused by intrinsic superconductivity in NiTe₂. In the revised manuscript, we have emphasized that NiTe₂ is not superconducting and added this reference (**Modification #1**). Furthermore, similar NiTe₂-based Josephson junctions were also reported by B. Pal *et al.* [Nat. Phys. 18, 1228-1233 (2022)] (Ref. 45) at 20 mK, which mainly focused on the Josephson diode effect.

Modification #1: We added two sentences in paragraph 3, page 3, which read “Note that NiTe₂ is not superconducting down to 30 mK³⁸. Since device D1 shown below was

measured around 70 mK, our observations should not be caused by intrinsic superconductivity in NiTe₂.”.

2. In fig 2, where the authors claimed the B_x can suppress the bulk supercurrent, my concern is the B_x is too large, even a tiny misalignment between the sample and B_x plane will lead to a considerable large magnetic field at z direction. For example, if there exists 5 degrees misalignment, 0.2 T B_x magnetic field will give rise to a ~ 17 mT B_z constant magnetic field. Therefore, the $+9$ mT B_z region shown in fig 2d is equivalent to the 12-26 mT region of fig 2a plus the influence of the remain real x direction magnetic field. And hence, due to the absence of the central peak, one can always obtain the edge dominated current profile by the method in supplementary section I.

Reply: We appreciate Reviewer #1 for raising this important question. We agree that the misalignment between the sample and in-plane magnetic field could induce the edge-dominated current profile. However, this is not the case in our work, and the misalignment has been corrected when an in-plane magnetic field is applied. We thank the reviewer for pointing out this issue, which helped us to notice that we did not clarify the correction of the misalignment in the previous version of our manuscript.

In fact, we can estimate the misalignment angle from the superconducting interference pattern (SIP) using the position of the central peak in B_z with finite $B_{x,y}$ applied. When an in-plane magnetic field is applied, the x -axis of the data shown in the main text has been corrected to eliminate the B_z component induced by $B_{x,y}$. For example, Fig. R1a shows the original data on D1 at $B_x = 0.1$ T, which indicates a shift of the central peak of around 0.5 mT, i.e., the B_z component is ~ 0.5 mT. Therefore, the misalignment angle should be close to 0.3 degree, and the B_z component at $B_x = 0.2$ T is ~ 1 mT, which can be corrected. The x -axis of the 2D color plots shown in Figs. 2c, 2d, 3b, 3d, 4b-e, and 5b has been corrected using this approach, and some more examples are shown in Fig. R1. In order to clarify this issue, we added Fig. R1 in the revised Supplementary Information and explained the correction of the misalignment (**Modification #2**).

Fig. R1. The original SIPs for different devices at finite B_x without eliminating the B_z component. The two dashed lines indicate the B_z component due to the misalignment.

Modification #2: We added Section II in the Supplementary Information, which includes Fig. R1 (Fig. S2) and the explanation of the correction of the misalignment.

3. The SIP in a JJ is largely dependent on the direction of the magnetic field and the shape of the sample, as discussed/observed by many previous works (Nat Nanotechnol 17, 39-44 (2022), J. Appl. Phys. 103, 07C707 (2008), Phys. Rev. B 107 094522 (2023)). Both the heights and periods of the lobes will be influenced by the magnetic field along another direction. Therefore, by applying a constant magnetic field along one direction, and measuring SIP with magnetic field along another direction is not suitable to extract the current profile by the method described in Supplementary Information Section I.

Reply: We agree with Reviewer #1 that “the SIP in a JJ is largely dependent on the direction of the magnetic field and the shape of the sample”. However, we respectfully disagree with that “by applying a constant magnetic field along one direction, and measuring SIP with magnetic field along another direction is not suitable to extract the current profile by the method described in Supplementary section I”. We thank the

reviewer for recommending previous works and would like to respectively emphasize that the mechanisms in these works are not suitable for our observations. **In our work, the applied magnetic field B_x is along the Josephson current direction, and then the current profile is extracted from the B_z -dependent interference patterns. Note that B_x does not contribute to the phase in the integration of the current density at each point over the junction area, and therefore the Dynes-Fulton approach is suitable for our regular-shaped junctions.** Whereas in these references, the shape dependence results from the perpendicular components regard to the current direction of the magnetic field along another direction, i.e., contributing to the phase in the integration of the current density. Even in this case, these references do not violate the Dynes-Fulton approach based on a careful treatment, especially the latter two. For example, the third reference recommended by the reviewer, Phys. Rev. B 107 094522 (2023), applies the Dynes-Fulton approach to analyze their interference patterns (Ref. 18 in this work) by dividing their junctions into slices because of the irregular shapes. In addition, a method similar to ours was also used in arXiv: 2211.01020 [Nano Lett. 23, 6102–6108 (2023)] under the in-plane magnetic field, which was recommended by Reviewer #3.

In Nat. Nanotechnol. 17, 39-44 (2022), the scope is the Josephson diode effect. The pronounced change of the SIP in this work is along B_y which is perpendicular to the current, whereas in our work it is along B_x which is parallel to the current. In this reference, when B_x is applied, “besides some variations of the apparent period of the patterns with B_x , **the lobe structure is close to the standard Fraunhofer shape**”. On the other hand, the significant influence from the magnetic field B_y perpendicular to the current in this reference is attributed to the supercurrent rectification (i.e., the Josephson diode effect) associated with the interplay of spin–orbit coupling and magnetic field. However, the Josephson diode effect is not the scope of our work.

In J. Appl. Phys. 103, 07C707 (2008), the Josephson junction is sandwich-like, while in our work it is planar. Therefore, the definitions of B_x and B_y are very different. The

most important effect of the magnetic field in this reference is the position-dependent phase modulation in the junction, which requires **the magnetic field to be perpendicular to the current** (junction). The critical current is an integral of the modulated current density at each point of the whole junction, a direct application of the Josephson equations, similar to the well-known Fraunhofer pattern for regular Josephson junctions. What this reference studied is the application to junctions of different shapes, including square-, triangular-, hexagonal-, and circular-shaped junctions. Of course, as the reviewer pointed out, the SIPs depend on the shape of the junction because of the integration over the whole cross-section of the junction. For these junctions, the Dynes-Fulton approach is still applicable if we can divide the particular shape into regular slices as the third recommended reference [Phys. Rev. B 107 094522 (2023)] did. For our regular junctions, this also holds but will be simplified directly to the Dynes-Fulton approach. Especially, we studied the effect of B_x which makes the usage of the Dynes-Fulton approach straightforward. Therefore, this reference and our work focus on different aspects, and the particular shape dependence in this reference does not apply to our observations. Since both works utilize a direct application of the Josephson equations to consider the critical current, they do not violate the Dynes-Fulton approach in principle.

In Phys. Rev. B 107 094522 (2023), the idea is similar to J. Appl. Phys. 103, 07C707 (2008) explained above. Irregular interference patterns are observed depending on the interface shape of the junctions, where irregular shapes are obtained due to the stack of two NbSe₂ flakes. Please note that the junction is perpendicular to the plane of the flakes because of the stacking, and the so-called in-plane magnetic field in this work is actually perpendicular to the current/junction, again, different from our case. In addition, the effective interface shape in our Josephson junctions are always rectangular-like, which are not expected to produce the irregular SIPs if the current is uniform. Furthermore, this reference also applies Dynes-Fulton approach to analyze the interference patterns (Ref. 18 in this work), by dividing their junctions into slices because of the irregular shapes, which exactly supports the validity of this method for our work.

We thank the reviewer again for recommending these references. To summarize, these references and our work focus on different aspects of Josephson junctions. The shape dependence of the SIPs comes from the integration of the current density over the whole junction. In principle, these references do not violate the Dynes-Fulton approach if we divide the junction into slices for specific junction shapes. Especially, in our work, applying a constant magnetic field along B_x only modifies the total intensity of the Josephson supercurrent and does not act on the phase at each point over the junction area. The total critical current of the junction with/without magnetic fields follows the Josephson equations, and therefore, the SIPs along B_z in our work can still be analyzed by the Dynes-Fulton approach as described in Supplementary Information Section I.

4. The authors mentioned that the I_c should show a $1/|B_z|$ global decay in page 6 and supplementary IV, and extracted the boundary I_c for different lobes, and hence claimed the observation of the hinge state in NiTe₂. The authors should explain it in more detail, and cite references. I cannot fully judge anything here without further explanation and detailed equations.

Reply: We thank Reviewer #1 for giving this valuable suggestion. In the last paragraph on page 6 of the initial manuscript, we wrote “If there is a SPE on the side surfaces, the damping of the critical boundary supercurrent (boundary- I_c) under B_y follows the Fraunhofer-like curve. Note that the $1/|B_z|$ global decay, which is a manifestation of the cancellation of the positive and negative supercurrent, helps to suppress the contribution of the bulk supercurrent for the side lobes¹¹.”

Let’s explain the $1/|B_z|$ global decay first. For a Josephson junction, the expression of the Fraunhofer curve is $I_c(B_z)/I_c(B_z = 0 \text{ T}) = |\sin(\pi\Phi/\Phi_0)/(\pi\Phi/\Phi_0)|$. If we look at the center of each side lobe, $|B_z|$ satisfies $|B_z|S/\Phi_0 = |n| + 1/2$, where S is the cross-sectional area. So, the height of the side lobes is $I_c(n)/I_c(B_z = 0 \text{ T}) = 1/(\pi(|n| + 1/2))$, which follows the $1/|B_z|$ global decay. Please note that such decay applies to the bulk state when B_z is applied, and also to the side surfaces (if exist) when

B_y is applied. Specifically, for the latter, assuming that the current flows through the side surfaces, we expect a critical boundary supercurrent (boundary- I_c) following the Fraunhofer-like curve under B_y which is perpendicular to the rectangular side surfaces (with $1/|B_y|$ global decay).

Next, we interpret the clue of hinge state by comparing the assumption of side surfaces and the experimental data. Our main explanation is that the experimental critical boundary supercurrent (boundary- I_c) at finite B_y is much larger than the calculated values if assuming a side-surface supercurrent, and hence it points to the hinge supercurrent.

Firstly, when $|B_z|$ is applied, the bulk supercurrent will follow the Fraunhofer curve, i.e., the $1/|B_z|$ global decay. Therefore, a strong suppression of the bulk supercurrent can be realized at the side lobes (a manifestation of the partial cancellation of the positive and negative supercurrent), highlighting the boundary supercurrent. Of course, the larger the serial number $|n|$ of the side lobes is, the heavier the suppression of the bulk supercurrent. As a result, the experimental boundary- I_c could be extracted approximatively from the height of side lobes (with $1/|B_z|$ suppression of the bulk supercurrent) in Fig. 2a and S5 for device D1, at $B_y = 0$ T and $B_y = 0.2$ T, respectively. The extracted boundary- I_c is shown in Fig. 5c.

Secondly, if there is a superconducting proximity effect on the side surfaces, we expect a Fraunhofer-like curve $I_c(B_y)/I_c(B_y = 0 \text{ T}) = |\sin(\pi\Phi/\Phi_0)/(\pi\Phi/\Phi_0)|$ in B_y perpendicular to the side surface. So, we can calculate the theoretical boundary- I_c at $B_y = 0.2$ T, as shown by the pink triangles in Fig. 5c. However, experimental boundary- $I_c(B_y = 0.2 \text{ T})$ (blue squares) are much larger than the calculated values (pink triangles), as shown in Fig. 5c, which is in contrast with a side-surface supercurrent. Furthermore, boundary- $I_c(B_y = 0.2 \text{ T})$ are comparable to the boundary- $I_c(B_x = 0.2 \text{ T})$ (red circles), which does not support the side-surface scenario, either,

because a large suppression of the supercurrent caused by the Fraunhofer-like decay is expected for B_y , but not for B_x .

Modification #3: To present a better explanation, in the revised version we have rewritten this part in the main text and Supplementary Information as shown below.

Starting from paragraph 2, page 7 of the main text, “Next, we interpret the clue of hinge states by comparing the assumption of side surfaces and the experimental data. Our main explanation is that the experimental critical boundary supercurrent (boundary- I_c) at finite B_y is much larger than the calculated values if assuming a side-surface supercurrent, and hence it points to the hinge supercurrent.

For a Josephson junction, the expression of the Fraunhofer curve is $I_c(B_z)/I_c(B_z = 0 \text{ T}) = |\sin(\pi\Phi/\Phi_0)/(\pi\Phi/\Phi_0)|$. If we look at the center of each side lobe, $|B_z|$ satisfies $|B_z|S/\Phi_0 = |n| + 1/2$, where S is the cross-sectional area. So the height of the side lobes is $I_c(n)/I_c(B_z = 0 \text{ T}) = 1/(\pi(|n| + 1/2))$, which follows the $1/|B_z|$ global decay¹¹. Note that the $1/|B_z|$ global decay, which is a manifestation of the partial cancellation of the positive and negative supercurrent, helps to suppress the contribution of the bulk supercurrent for the side lobes. Such decay applies to the bulk state when B_z is applied, and also to the side surfaces (if exist) when B_y is applied. Specifically, for the latter, assuming that the current flows through the side surfaces, we expect a boundary- I_c following the Fraunhofer-like curve under B_y , which is perpendicular to the rectangular side surfaces (with $1/|B_y|$ global decay).

When $|B_z|$ is applied, the bulk supercurrent will follow the Fraunhofer curve, i.e., the $1/|B_z|$ global decay. Therefore, a strong suppression of the bulk supercurrent can be realized at the side lobes, highlighting the boundary supercurrent. Of course, the larger the serial number $|n|$ of the side lobes is, the heavier the suppression of the bulk supercurrent. As a result, the experimental boundary- I_c could be extracted approximatively from the height of side lobes (with $1/|B_z|$ suppression of the bulk

supercurrent) in Fig. 2a and S5 for device D1, at $B_y = 0$ T and $B_y = 0.2$ T, respectively. The extracted boundary- I_c is shown in Fig. 5c.

If there is a superconducting proximity effect on the side surfaces, we expect a Fraunhofer-like curve $I_c(B_y)/I_c(B_y = 0 \text{ T}) = |\sin(\pi\Phi/\Phi_0)/(\pi\Phi/\Phi_0)|$ in B_y perpendicular to the side surface. Considering a nanoplate thickness of ~ 30 nm, and a separation of the electrodes ~ 300 nm, theoretical boundary- I_c at $B_y = 0.2$ T could be calculated, as shown by the pink triangles in Fig. 5c (detailed analysis is shown in Supplementary Section V). However, experimental boundary- $I_c(B_y = 0.2 \text{ T})$ (blue squares) are much larger than the calculated values (pink triangles), which is in contrast with a side-surface supercurrent. Furthermore, boundary- $I_c(B_y = 0.2 \text{ T})$ are comparable to the boundary- $I_c(B_x = 0.2 \text{ T})$ (red circles), which does not support the side-surface scenario, either, because a large suppression of the supercurrent caused by the Fraunhofer-like decay is expected for B_y , but not for B_x .”

In the Supplementary Information Section V, “As we mentioned in the main text, if the boundary supercurrent originates from the side surfaces (rather than the hinges), we can calculate its value at a certain B_y through the Fraunhofer-like decay curve $I_c(B_y)/I_c(B_y = 0 \text{ T}) = |\sin(\pi\Phi/\Phi_0)/(\pi\Phi/\Phi_0)|$. To do so, we need to suppress the contribution of the bulk supercurrent, which can be achieved by applying a small B_z to render the Fraunhofer-like decay of the bulk supercurrent itself. Therefore, we inspect the side lobes of the SIP in Fig. 2a in the main text, where the bulk supercurrent has been suppressed due to the Fraunhofer-like decay in B_z . We assign the height of the side lobes (excluding the central lobe) of the SIP in Fig. 2a as I_c of the side-surface, i.e., side-surface- $I_c(B_y = 0 \text{ T})$. Using the equation $I_c(B_y)/I_c(B_y = 0 \text{ T}) = |\sin(\pi\Phi/\Phi_0)/(\pi\Phi/\Phi_0)|$, we can calculate the side-surface- I_c at $B_y = 0.2$ T. In the main text, we assume that the effective junction length of the side surface is comparable to the separation of the electrodes, i.e., ~ 300 nm. In fact, it could be underestimated if considering the flux focusing effect of the electrodes. We thus test several conditions here, as shown in Fig.

S4, and side-surface- I_c at $B_y = 0.2$ T is always smaller than the height of the first side lobe, which is $\sim 0.21I_c(B_y = 0$ T). However, the experimental boundary- $I_c(B_y = 0.2$ T) is much larger than $0.21I_c(B_y = 0$ T) as shown in Fig. 5c in the main text, in contrast to the side-surface scenario.”

5. In line 157, the authors claimed the bulk supercurrent behavior with Fig. 5a and 5b. But with only one lobe shown in the SIPs, one can hardly say it's from the Fraunhofer pattern or SQUID pattern. Better SIPs should be provided to support the claim.

Reply: We thank Reviewer #1 for raising this issue. Firstly, let us compare the Fraunhofer pattern and the SQUID pattern. As shown in Fig. 2e, there will be a fast decay ($1/|B_z|$) for the Fraunhofer pattern and usually only a few side lobes can be observed, while the SQUID pattern shows a much slower decay and many side lobes can be measured. The only one central lobe in the SIPs of Figs. 5a and 5b is merely possible for the Fraunhofer case in our measurement. Why there is only one central lobe but not side lobes? The reason is that for the junction shown in the left inset of Fig. 5a, the supercurrent extends towards the outside of the junction area due to the superconducting proximity effect, and therefore the current density is not uniform, as shown roughly by the right inset of Fig. 5a. So, the SIP for this case is distorted from the standard Fraunhofer pattern and side lobes can be hardly observed. In this case, we sincerely feel regretful that we did not obtain better SIPs, which we think is primarily determined by the geometry. In addition, a similar pattern was reported in the Supplementary Information of Ref. 15 [Nat. Mater. 19, 974-979, (2020)].

Modification #4: In paragraph 3, page 6, to avoid confusion we have added two sentences which read “The SIP only presents a central lobe as depicted in Fig. 5a, which is merely possible for the Fraunhofer case and could be attributed to the supercurrent extending towards the outside of the junction area due to the superconducting proximity effect. The current density is not uniform as shown by the right inset of Fig. 5a, and thus the SIP is distorted from the standard Fraunhofer pattern and side lobes can be

hardly observed.”

6. I noticed a B_x arrow at the rightmost position in Fig. 2, what does that mean? And there is a white square in Fig. 6b which covers some band structures.

Reply: We appreciate Reviewer #1 for pointing out these typos. The B_x arrow means the increase of magnetic field B_x from top to bottom. We have elucidated it in the caption of Fig. 2 (**Modification #5**). The white square was a typo when we plotted and merged the figures, and we have removed it from Fig. 6b (**Modification #6**).

Modification #5: We have elucidated the arrow in Fig. 2e by writing “The B_x arrow means the increase of magnetic field B_x from top to bottom.” in the caption.

Modification #6: We removed the white square from Fig. 6b which was a typo.

Reviewer #2

This experimental work discusses the measurement of Type-II Dirac semimetal NiTe₂-based Josephson junction. Authors investigated magnetic field interference pattern of Josephson junctions and suggest that Type-II Dirac semimetal NiTe₂ have hinge states, which mediate supercurrent. When a perpendicular magnetic field is applied to the Josephson junction, the magnetic field interference pattern (Fraunhofer pattern) shows slowly vanishing behavior, suggesting an edge-enhanced current. When an in-plane magnetic field is added, the Fraunhofer pattern is transformed into that of a superconducting quantum interference device (SQUID). Analysis of the magnetic field interference pattern and its Josephson current density shows an unconventional interference pattern and claims the existence of hinge states in NiTe₂.

Reply: We sincerely appreciate Reviewer #2 for the thorough examination of our

manuscript and for providing valuable comments and suggestions, which significantly improves our manuscript. Enclosed below are our detailed responses.

1. Authors mentioned “the magnetic field filtering of the supercurrent functions as a compelling route to acquire the nearly pure boundary supercurrent in topological/unconventional materials based JJs.” in lines 64-67. Could authors explain why and how that “magnetic field filtering” happens in detail? Is there any reference for this mechanism?

Reply: We sincerely appreciate Reviewer #2 for the valuable inquiry regarding “magnetic field filtering”, which was also mentioned in Comment 2 below. We firmly believe that the key mechanism behind “magnetic field filtering” lies in the spin-momentum locking of the hinge states, as elaborated in pages 9 and 10 of the manuscript. It is crucial to note that the magnetic field filtering of the supercurrent is effective only when the magnetic field is oriented along the Josephson current, denoted as B_x .

Fig. R2. Schematic illustration of the spin distribution. The spin in the bulk is randomized, while on the hinges it is locked to be in the plane perpendicular to the hinges.

In the case of planar Josephson junctions, when the magnetic field is parallel to the current, the Fraunhofer-like decay of the supercurrent is absent (please refer to the reply

to Comment 4 of Reviewer #1 and **Modification #3** for the explanation of the Fraunhofer-like decay). Considering the small thickness of NiTe₂ plates, orbital pair-breaking effect is relatively weak at such small magnetic fields in our work. As a result, the Zeeman effect should dominate the suppression of the supercurrent.

Then, we consider the symmetries on the hinges (in x direction): mirror symmetry (M_x) and time reversal (T). The M_x makes the spin orientation of hinge states satisfy $\sigma_x(k) = \sigma_x(-k)$, while T makes $\sigma_x(k) = -\sigma_x(-k)$. Thus, the combined symmetry TM_x yields $\sigma_x(k) = 0$ for the non-degenerate hinge states. In other words, **due to the co-existence of M_x and T , the electron spins of the hinge states are locked to be in the plane perpendicular to the hinges (along the x direction, i.e., B_x , as illustrated in Fig. R2 and Fig. 6h of the revised manuscript)**. Consequently, this locking protects the Cooper pairs of the hinge states from undergoing depairing in B_x . In contrary, the spins of the bulk states are randomized without such protection. As a result, the coupling between spins and B_x (Zeeman energy) for the bulk is much larger than that for the hinges, and therefore the Cooper pairs of the bulk are easier to break. A similar mechanism has been previously discussed in Ising superconductors to explain the giant upper critical field [Science 350, 1353-1357 (2015); Nat. Phys. 12, 144-149 (2016)]. Due to the spin-orbit-coupling locked spin perpendicular to the MoS₂ film, an in-plane upper critical field much larger than the Pauli limit was observed. In addition, we note that the reference [arXiv: 2211.01020 or Nano Lett. 23, 6102–6108 (2023)] recommended by Reviewer #3 in Comment 2 reports the narrow superconducting channels induced by the in-plane magnetic field in NbSe₂-graphene-NbSe₂ Josephson junctions, which is associated with the Ising superconductivity in NbSe₂. This work could also be conceptualized as a form of magnetic filtering of supercurrent.

2. Authors mentioned, “the magnetic field filtering of supercurrent is absent for B_y .” Could authors explain why magnetic field filtering for B_y is ineffective and how this relates to the magnetic field filtering effect?

Reply: We follow the reply to the last comment. B_y represents the magnetic field perpendicular to the current. In this case, the acquired Zeeman energy to break Cooper pairs for the spins of the hinges is almost the same as that for the bulk. As a result, the Cooper pairs are not protected by the spin-momentum locking. Therefore, the magnetic field filtering of supercurrent is absent for B_y . In order to explain the magnetic field filtering in more detail, we have rewritten this part in the revised manuscript.

Modification #7: We have rewritten this part starting from paragraph 2, page 9, added Fig. R2 as Fig. 6h, and added references. “Then, we consider the symmetries on the hinges (in x direction): mirror symmetry (M_x) and time reversal (T). The M_x makes the spin orientation of hinge states satisfy $\sigma_x(k) = \sigma_x(-k)$, while T makes $\sigma_x(k) = -\sigma_x(-k)$. Thus, the combined symmetry TM_x yields $\sigma_x(k) = 0$ for the non-degenerate hinge states. In other words, due to the co-existence of M_x and T , the electron spins of the hinge states are locked to be in the plane perpendicular to the hinges, as illustrated in Fig. 6h. The spin orientation of the non-degenerate hinge states aligns well with the filter effect of the supercurrent under B_x , as explained below.

In the case of planar Josephson junctions, when the magnetic field is parallel to the current, the Fraunhofer-like decay of the supercurrent is absent. Considering the small thickness of NiTe₂ plates, orbital pair-breaking effect is relatively weak at such small magnetic fields. Therefore, the Zeeman effect should dominate the suppression of the supercurrent. Consequently, the locked spin protects the Cooper pairs of the hinge states from undergoing depairing in B_x . In contrary, the spins of the bulk states are randomized without such protection. The coupling between spins and B_x (Zeeman energy) for the bulk is much larger than that for the hinges, and therefore the Cooper pairs of the bulk are easier to break. A similar mechanism has been previously discussed in Ising superconductors to explain the giant upper critical field^{49, 50}. Due to the spin-orbit-coupling locked spin perpendicular to the MoS₂ film, an in-plane upper critical field much larger than the Pauli limit was observed.

Regarding B_y , the acquired Zeeman energy to break Cooper pairs for the spins of the hinges is almost the same as that for the bulk. Hence, the Cooper pairs are not protected by the spin-momentum locking. Therefore, the magnetic field filtering of supercurrent is absent for B_y . (Our main train of thought on the hinge states and the filtering effect is shown in Supplementary Section IX.) Recently, modulation of supercurrent induced by the in-plane magnetic field was also observed in graphene systems, which could also be conceptualized as a form of magnetic filtering of supercurrent^{51, 52.}”

3. There are at least three relevant works, including [Nature Phys 13, 87–93 (2017)], [Nat Commun 9, 3478 (2018)], and [Nat. Phys. 18, 1228–1233 (2022)]. Especially, [Nat. Phys. 18, 1228–1233 (2022)] have already investigated the magnetic field interference with an in-plane magnetic field (B_x) parallel to the Josephson current and with the perpendicular magnetic field (B_z), like the work by authors. I wonder how authors’ works are significantly different from those previous works.

Reply: We appreciate Reviewer #2 for recommending these interesting references. After carefully scrutinizing these works, we realized that they mainly focused on the finite-momentum Cooper pairing (or related Josephson diode effect) induced by the in-plane magnetic field together with spin-orbit coupling or spin polarized surface states. There are two main features for finite-momentum Cooper pairing. One is the disappearance of superconducting interference at some special B_x values [Nature Phys 13, 87–93 (2017)]. Another is the critical current maxima occur at increasingly large values of $|B_z|$ as B_x grows [Nat Commun 9, 3478 (2018)] and [Nat. Phys. 18, 1228–1233 (2022)]. Both of them are absent in our work. In the revised manuscript, we have cited these references as Refs. 43-45 (**Modification #8**).

In Nat. Phys. 13, 87–93 (2017), finite-momentum Cooper pairing was observed in HgTe quantum wells, a manifestation of the interplay between in-plane magnetic field and spin-orbit coupling. Please note that in this work B_x denotes the in-plane magnetic field perpendicular to the Josephson current, which is different from our work. The induced

finite Cooper pair momentum in HgTe requires structural inversion asymmetry or bulk inversion asymmetry (spin-orbit coupling).

In Nat. Commun. 9, 3478 (2018) and Nat. Phys. 18, 1228–1233 (2022), the finite-momentum Cooper pairing is attributed to the spin-polarized surface states in Bi₂Se₃ and NiTe₂, respectively. The main feature in these references is the critical current maxima occur at increasingly large values of $|B_z|$ as B_x grows. Furthermore, in Nat. Phys. 18, 1228–1233 (2022), the authors mainly studied the Josephson diode effect due to the finite-momentum Cooper pairing and did not focus on the superconducting interference pattern at fixed B_x .

In our work, we mainly focused on the superconducting interference pattern at fixed B_x , which requires an in-plane magnetic field parallel to the current. We mainly observed the magnetic filtering of the hinge supercurrent. Our detailed calculations showed that NiTe₂ has the unconventional nature of charge mismatch, which gives rise to the obstructed hinge states. The locked spin perpendicular to the hinges contributes to such filtering. The incorporation of our experimental and theoretical results supports the existence and the filtering of the hinge supercurrent.

Reviewer #3

The manuscript by Le and co-authors reports on measurements of Josephson devices consisting of NbTiN superconducting contacts and exfoliated NiTe₂ flakes as weak links. The authors use standard out-of-plane interference patterns to map the current distribution within the junction. The main result of this work is found upon application of very small magnetic fields directed in the plane of the sample. Fields applied parallel to the current flow, in the range of < 0.1 T, cause the supercurrent to re-arrange such that the zero-field Fraunhofer pattern disappears, giving rise to a SQUID-like oscillation. The latter indicates that current is being carried along the edges – an effect

which the authors associate with the presence of boundary states. In this sense, the in-plane magnetic field is being used as a way to filter between distinct carrier channels – with the edge channel being immune to its effects.

The authors argue that the origin of the boundary supercurrent is in fact due to hinges, rather than side edges, by presenting the dependence on the field B_y , oriented perpendicular to the current propagation direction. They finally present a set of calculations, suggesting the existence of hinge-modes of obstructed-Wannier orbital type.

Reply: We sincerely thank Reviewer #3 for the very careful review of our manuscript and for providing valuable comments and suggestions on our work. The very insightful feedback has significantly contributed to the improvement of our manuscript. Please find below our detailed responses to each of the comments.

This manuscript is very well written, it is clear, and I find it quite convincing. If indeed the authors observe hinge modes, this would be a nice advance over existing state of the art. If the authors can answer the following points, this work can be considered for publication in Nature Communications.

Reply: We are grateful to Reviewer #3 for these positive comments and for the recommendation of publication.

1. My main concern is related to the proof of transport being carried by hinge modes. There are, by now, several works which report supercurrent transport at edges, even such which are not topological, and the transition from bulk to edge due to applied in-plane field is by no means a proof the presence of special edge modes. Edge current can be related to edge accumulation or other effects – e.g. in graphene (Allen et al., Ref 20, and Zhu et al., Nature Comm. 8 14552 (2017), which should be cited here), and also in $\text{Bi}_2\text{O}_2\text{Se}$ (Ref 21).

Reply: We appreciate Reviewer #3 for raising this important question. Indeed, the proof of transport being carried by hinge states is essential.

We agree with Reviewer #3 that “several works which report supercurrent transport at edges, even such which are not topological”, and Nature Comm. 8 14552 (2017) has been cited in the revised manuscript (Ref. 23). These works mentioned by the reviewer primarily focused on the edge supercurrent tuned by the gate voltage in materials with low carrier density. Indeed, “edge current can be related to edge accumulation or other effects”, and “the transition from bulk to edge due to applied in-plane field is by no means a proof the presence of special edge modes”. It is even more difficult to tell if the edge is the side surface or the hinge.

Fig. R3. Train of thought on the existence of hinge states.

How we achieved the explanation of the existence of hinge states is incorporating our experimental observations with the theoretical calculations. To show an overview of our train of thought, we summarized the results roughly in Fig. R3.

Experimentally, we observed the transition from the bulk to the edge when an in-plane magnetic field (B_x) parallel to the current is applied, which we call the filtering effect. However, in-plane magnetic field (B_y) perpendicular to the current does not show such effect. To find the clue on the origin of the edge state, i.e., the side surface or the hinge, we assume that it is the side surface and compare with the experimental data. But we

find contrary between the experimental boundary- I_c at $B_y = 0.2$ T and the side-surface scenario, as shown in Fig. 5c. (Please refer to our reply to Comment 4 of Reviewer #1 where we explained such contrary in more detail and we revised the manuscript accordingly to reach a better presentation, i.e., **Modification #3**.) Therefore, it points to the hinge supercurrent.

Theoretically, our detailed calculations show that NiTe₂ has the unconventional nature of charge mismatch, which gives rise to the obstructed hinge states. Importantly, due to the co-existence of time reversal symmetry and mirror symmetry, the electron spins of the hinge states are locked to be in the plane perpendicular to the hinges (along the x direction), which could explain the filter effect of the supercurrent under B_x . As illustrated in Fig. R2 shown above and Fig. 6h of the revised manuscript, this locking protects the Cooper pairs of the hinge states from undergoing depairing in B_x . In contrary, the spins of the bulk states are randomized without such protection, and hence the filtering of the hinge supercurrent can be observed. B_y represents the magnetic field perpendicular to the current. In this case, the acquired Zeeman energy to break Cooper pairs for the spins of the hinges is almost the same as that for the bulk. As a result, the Cooper pairs are not protected by the spin-momentum locking. Therefore, the magnetic field filtering of supercurrent is absent for B_y .

Therefore, combining the experimental and theoretical results, we reached a self-consistent conclusion that the supercurrent is carried by hinge states. On the other hand, the side-surface scenario contradicts with both the experimental data and the theoretical calculations, and we do not know how to explain the filtering effect merely based on the edge accumulation or other effects, like in graphene or Bi₂O₂Se systems.

Modification #9: In order to present our logic and train of thought, we have added Fig. R3 and the explanation to the Supplementary Information Section IX.

2. A field-induced transition of supercurrent to a narrow channel (at the edge or not) may also seen in graphene: See e.g. PRB 103, 115401 (2021), and in arxiv: 2211.01020.

There works too, I believe, should be cited here.

Reply: We appreciate Reviewer #3 for recommending these two intriguing references and we have duly cited them in the revised manuscript (Refs. 51 and 52). These works provide valuable insights into the field-induced transition of supercurrent to a narrow channel in graphene. The former was attributed to the unique effect of ripples in an atomically thin graphene layer. As for the latter, the main mechanism remained unclear, but the authors mentioned the crucial role of Ising spin-orbit coupling, which aligns with our explanation associated with the spin-momentum locking of hinge states, as we response to Reviewer #2's Comments 1&2.

3. There should be 2 hinges on each side – giving rise to a beating effect. Is there any indication for such beating?

Reply: We thank Reviewer #3 for putting forward this question. Regrettably, we did not observe this effect in our work, despite measuring more than 10 devices. We think it might be attributed to the superconducting interference patterns persisting only to several side lobes. We genuinely appreciate the reviewer for raising this point, and this is a worthy direction for hinge states.

4-1. The reason why bulk states disappear at such low in-plane field is not properly explained.

Reply: We appreciate Reviewer #3 for raising this important question regarding the low in-plane magnetic field to suppress the bulk supercurrent, which we did not explain in the previous manuscript. This can be interpreted by the Gaussian-like decay of the bulk supercurrent.

The behavior of the supercurrent of SNS Josephson junctions in an in-plane magnetic field has been studied. According to Phys. Rev. B 76, 064514 (2007), the critical current I_c for bulk states shows Gaussian-like decay, given by $I_c(B_x) = I_c(0)e^{-0.145\Gamma_{sf}/\epsilon T} \approx$

$I_c(0)e^{-B^2/2\sigma^2}$. Here Γ_{sf} is the spin-flip scattering, ϵ_T is the Thouless energy and σ is the effective decay coefficient. When the spin-flip length $L_{sf} = \sqrt{\hbar D/2\Gamma_{sf}}$ (D is the diffusion coefficient in the normal metal) becomes smaller than the length L of the SNS junction (L could be the separation of the electrodes), spin-flip scattering could act as a pair-breaking mechanism for the Cooper pairs, dominating the decay of supercurrent. In such case, Γ_{sf} is proportional to the square of magnetic field B^2 . Based on this scenario, it is possible for the supercurrent on bulk states to vanish at low in-plane fields. We noticed that the critical in-plane field H_{c2}^{\parallel} of the bulk states in Cd₃As₂-based planar Josephson junctions is also small, which is around 0.1 T in Nat. Commun. 11, 1150 (2020).

Modification #10: We have added the detailed explanation of the low in-plane field to suppress the bulk supercurrent in the Supplementary Information Section III.

4-2. What drives the difference between individual samples? I don't really understand why the fabrication process should have any role, as suggested in the Supplementary.

Reply: We thank Reviewer #3 for putting forward this issue. As we mentioned in the Supplementary Information Section III, devices D2, D3 and D4 were fabricated simultaneously, and they showed similar in-plane magnetic field for suppressing the bulk supercurrent. In fact, more than 7 devices were fabricated at that time and none of them exhibited any pronounced difference in the critical in-plane field. However, devices D1 and D5 showing a larger critical in-plane field were fabricated in another batch several months earlier. Therefore, we think that the difference of the critical in-plane field depends on the detailed fabrication process, such as the quality of superconducting films and the cleanness of the interface. These factors will affect the quality of the superconducting proximity effect, and thus the critical supercurrent and magnetic field.

5. On page 6, better clarify from the 1D hinges, rather than from the 2D vertical side-

surfaces.

Reply: We thank Reviewer #3 for this suggestion. Indeed, it would be better to clarify directly from the 1D hinges. However, we found that the description flow of the experimental data and the theoretical calculation, especially Fig. 5c, is more straightforward by assuming the 2D side surface scenario. Since Reviewer #1's Comment 4 also raised questions on this part, we noticed that a better presentation is required. We have revised this paragraph to present a more detailed and clearer interpretation. Please refer to **Modification #3** for a detailed description.

6. Line 32 – “futile” bulk contributions. I am not sure the work “futile” is the best choice here.

Reply: We thank Reviewer #3 for pointing out this improper wording.

Modification #11: We have deleted the word “futile” in line 32 in the revised manuscript.

7. Line 37 – “which is a well-known bulk-boundary correspondence” – this entire first sentence does not make grammatical sense.

Reply: We thank Reviewer #3 for pointing out this error. We have rewritten the first sentence in paragraph 2, page 2 as “Bulk band topology permeates in three- and two-dimensional (3D and 2D) condensed matter, e.g., topological insulators and topological semimetals, and gives rise to gapless surface/edge states. These states result from the well-known bulk-boundary correspondence and are topologically protected¹⁻³.”

(Modification #12)

8. Line 40 – “Josephson junctions have served as a powerful tool to reveal boundary states in topological materials” – citations 11-21. The authors should note that some of their citations are graphene systems, which are not topological. The text should be clear on this.

Reply: We appreciate Reviewer #3 for pointing out this typo. We have replaced “topological materials” with “topological materials¹²⁻²⁰ as well as non-topological materials²¹⁻²³”. **(Modification #13)**

9. Line 211 – the word “remanence” – I assume the authors intended to use “remnants”, or “a remnant of”.

Reply: We thank Reviewer #3 for pointing out this typo. We have replaced the “remanence” with the “remnants”. **(Modification #14)**

10. In Line 217, I didn’t understand what is the orientation of the locked spins – a diagram should be added.

Reply: We appreciate Reviewer #3 for this valuable suggestion. The locked spins are perpendicular to the hinges, namely B_x , and we have added a diagram as Fig. 6h (included in **Modification #7**). Please also refer to Fig. R2 in this reply.

11. The term “Obstructed hinge states” is used early on – I think this term should be defined in the text in the early part of the manuscript.

Reply: We thank Reviewer #3 for this suggestion. We modified “As a result of the mismatch between average electronic centers and atomic positions, the obstructed states emerge on the boundary,” in line 48 as “As a result of the mismatch between average electronic centers and atomic positions, the obstructed hinge/edge states emerge on the boundary,”. **(Modification #15)**

Reviewer #4

In this work, the authors fabricated a NiTe₂-based Josephson junction by coupling a NiTe thin film to two superconducting NbTiN electrodes. The Fraunhofer patterns of the Josephson junctions (critical current as a function of out-of-plane magnetic field) were observed under different conditions. The authors observed that:

1. Edge transport occurred even in the absence of an in-plane magnetic field, as shown in Fig. 2a and the current distribution in Fig. 2b.
2. The bulk supercurrent transport is suppressed by a magnetic field along the current direction, and the edge transport becomes more prominent, as shown in Figs. 2c and 2d.
3. From Figs. 3c and 3d, a small magnetic field of 0.04T is enough to suppress the bulk supercurrent, and the edge supercurrent can dominate the transport.
4. On the other hand, the effect of B_y is very different from that of B_x . A small B_y is not enough to suppress the bulk supercurrent, as shown in Figs. 4c to 4e.
5. The authors showed by DFT calculations that there are hinge states in NiTe₂.

Reply: We sincerely appreciate Reviewer #4 for the meticulous examination of our manuscript and providing valuable comments and suggestions on our work, which helped us a lot to improve the quality and clarity of our manuscript. Below, we present detailed responses to address each of the comments.

The experimental results of this work are quite nice.

Reply: We appreciate Reviewer #4 for the positive comments.

1. NiTe₂ is metallic and there are supposed to be a large number of bulk conducting channels. Why can the hinge states dominate the transport as shown in Fig. 2b? From

Fig. 2b, it is clear that there are significant edge contributions to the supercurrent.

Reply: We sincerely thank Reviewer #4 for raising this question. As discussed in Nat. Phys. 17, 542-546 (2021) (Ref. 12), superconducting interference is a convenient approach to distinguish supercurrent carried by a metallic state from another metallic one. The key ingredient is the different scattering/dephasing/transport mechanisms for these two states. Owing to the less dephasing and decoherence effects on hinge states due to various kinds of protection, the hinge supercurrent is expected to be more robust than that of the bulk states, even if the latter is also metallic. Indeed, hinge states of metallic compounds have been verified in Josephson junctions fabricated on Cd₃As₂, Bi, and WTe₂ (Refs. 12-17). In our work, the spin of the electrons on the hinges is locked to be perpendicular to the hinges. This locking protects the Cooper pairs of the hinge states from undergoing depairing, and therefore it is more robust. In contrary, the spins of bulk states are randomized without such protection. As a result, it is possible to observe pronounced hinge supercurrent.

2-1. Why can a small magnetic field along the supercurrent suppress the bulk supercurrent so much? This is not explained well in the manuscript. Instead of some handwaving arguments, can the authors provide a more rigorous theoretical support? The current level of reasoning is certainly not enough for Nature Communications.

Reply: We thank Reviewer #4 for raising this important question, as well as Reviewer #3's Comment 4-1. This question stimulated us to present a detailed explanation. We attribute the small in-plane magnetic field required for suppressing the bulk supercurrent to the Gaussian-like decay of the bulk supercurrent, which is caused by the spin-flip scattering when the magnetic field is parallel to the current [Phys. Rev. B 76, 064514 (2007)].

As we have shown in the response to Comment 4-1 of Reviewer #3, according to Phys. Rev. B 76, 064514 (2007), the critical current I_c for bulk states shows Gaussian-like decay, given by $I_c(B_x) = I_c(0)e^{-0.145\Gamma_{sf}/\epsilon_T} \approx I_c(0)e^{-B^2/2\sigma^2}$. Here Γ_{sf} is the spin-flip

scattering rate, ϵ_T is the Thouless energy and σ is the effective decay coefficient. When the spin-flip length $L_{sf} = \sqrt{\hbar D / 2\Gamma_{sf}}$ (D is the diffusion coefficient in the normal metal) becomes smaller than the length L of the SNS junction (L could be the separation of the electrodes), spin-flip scattering could act as a pair-breaking mechanism for the Cooper pairs, dominating the decay of supercurrent. In such case, Γ_{sf} is proportional to the square of magnetic field B^2 . Based on this scenario, it is possible for the supercurrent on bulk states to vanish at such low in-plane fields. We noticed that the critical in-plane field H_{c2}^{\parallel} of the bulk states in Cd₃As₂-based planar Josephson junctions is also small, which is around 0.1 T in Nat. Commun. 11, 1150 (2020). We have added this part in the revised Supplementary Information Section III.

(Modification #9)

2-2. I am not aware of similar effects of magnetic fields suppressing bulk supercurrents. Have similar effects been observed in other Josephson junctions?

Reply: As we mentioned in the last reply, bulk supercurrent suppressed under small in-plane magnetic field was also observed in Cd₃As₂-based planar Josephson junctions. The critical in-plane field is around 0.1 T in Nat. Commun. 11, 1150 (2020). Even though it is still larger than ours (~ 0.04 T), the reason could be attributed to the faster Gaussian-like decay in NiTe₂-based Josephson junctions, given by $I_c(B_x) = I_c(0)e^{-0.145\Gamma_{sf}/\epsilon_T} \approx I_c(0)e^{-B^2/2\sigma^2}$. As we reply to Comment 2-1, Γ_{sf} is proportional to the square of magnetic field B^2 . It implies that a faster decay behavior represents a smaller ϵ_T . The Thouless energy $\epsilon_T = \hbar D / L^2$, where D is the diffusion coefficient in the normal metal, and L is the length of the SNS junction. L is comparable between our junction and this reference's junction (L could be the separation of the electrodes ~ 300 nm). Therefore, if the diffusion coefficient in NiTe₂ is smaller than in Cd₃As₂, a faster Gaussian-like decay in NiTe₂-based Josephson junctions is possible.

3. Similarly, why is there such a large difference between the effects of B_y and B_x ? Can

the authors provide a more rigorous theory?

Reply: We appreciate Reviewer #4 for raising this important issue. As we mentioned in the revised manuscript (**Modification #7**) and the responses to the Comments 1&2 of Reviewer #2, we firmly believe that the key mechanism behind “magnetic field filtering” lies in the spin-momentum locking of hinge states. It is crucial to note that the magnetic field filtering of the supercurrent is effective only when the magnetic field is oriented along the Josephson current, denoted as B_x .

For the ease of reading, we repeat some of our responses above. In the case of planar Josephson junctions, when the magnetic field is parallel to the current, the Fraunhofer-like decay of the supercurrent is absent (please refer to the reply to Comment 4 of Reviewer #1 and **Modification #3** for the explanation of the Fraunhofer-like decay). Considering the small thickness of NiTe₂ plates, orbital pair-breaking effect is relatively weak at such small magnetic fields in our work. As a result, the Zeeman effect should dominate the suppression of the supercurrent.

Then, we consider the symmetries on the hinges (in x direction): mirror symmetry (M_x) and time reversal (T). The M_x makes the spin orientation of hinge states satisfy $\sigma_x(k) = \sigma_x(-k)$, while T makes $\sigma_x(k) = -\sigma_x(-k)$. Thus, the combined symmetry TM_x yields $\sigma_x(k) = 0$ for the non-degenerate hinge states. In other words, **due to the co-existence of M_x and T , the electron spins of the hinge states are locked to be in the plane perpendicular to the hinges (along the x direction, i.e., B_x , as illustrated in Fig. R2 and Fig. 6h of the revised manuscript)**. Consequently, this locking protects the Cooper pairs of the hinge states from undergoing depairing in B_x . In contrary, the spins of the bulk states are randomized without such protection. As a result, the coupling between spins and B_x (Zeeman energy) for the bulk is much larger than that for the hinges, and therefore the Cooper pairs of the bulk are easier to break. A similar mechanism has been previously discussed in Ising superconductors to explain the giant upper critical field [Science 350, 1353-1357 (2015). Nat. Phys. 12, 144-149 (2016)].

Due to the spin-orbit coupling locked spin perpendicular to the MoS₂ film, an in-plane upper critical field much larger than the Pauli limit was observed.

On the other hand, B_y represents the magnetic field perpendicular to the current. In this case, the acquired Zeeman energy to break Cooper pairs for the spins of the hinges is almost the same as that for the bulk. As a result, the Cooper pairs are not protected by the spin-momentum locking. Therefore, the magnetic field filtering of supercurrent is absent for B_y . In order to explain the magnetic field filtering in more detail, we have rewritten this part in the revised manuscript (**Modification #7**: we have rewritten paragraph 2, page 9, added Fig. R2 as Fig. 6h, and added references.).

4-1. From the Fraunhofer pattern and the current distribution, the authors can probably claim that there are edge states in NiTe₂. However, it is hard to identify them as hinge states. How can the authors rule out the possibility of having edge states?

Reply: We sincerely appreciate Reviewer #4 for raising this question regarding to distinguish between edge states and hinge states. It is obvious that our explanation about this part was not explicit enough in the initial manuscript, and the Reviewers #1 and #3 also asked similar questions. These crucial comments stimulated us to improve our interpretation and presentation, and we have rewritten this part in the revised manuscript (**Modification #3**). As we described in the present manuscript, the crucial point to rule out the edge states (i.e., side-surface states) is comparing the experimental values of boundary- I_c with the theoretical values if assuming the side-surface supercurrent. Our main explanation is that the experimental critical boundary supercurrent (boundary- I_c) at finite B_y is much larger than the calculated values and hence it points to the hinge supercurrent. To show an overview of our train of thought, we summarized the results roughly in Fig. R3.

Again, we repeat part of our responses. Experimentally, let's explain the $1/|B_z|$ global decay first. For a Josephson junction, the expression of the Fraunhofer curve is $I_c(B_z)/I_c(B_z = 0 \text{ T}) = |\sin(\pi\Phi/\Phi_0)/(\pi\Phi/\Phi_0)|$. If we look at the center of each side

lobe, $|B_z|$ satisfies $|B_z|S/\Phi_0 = |n| + 1/2$, where S is the cross-sectional area. So the height of the side lobes is $I_c(n)/I_c(B_z = 0 \text{ T}) = 1/(\pi(|n| + 1/2))$, which follows the $1/|B_z|$ global decay. Please note that such decay applies to the bulk state when B_z is applied, and also to the side surfaces (if exist) when B_y is applied. Specifically, for the latter, assuming that the current flows through the side surfaces, we expect a critical boundary supercurrent (boundary- I_c) following the Fraunhofer-like curve under B_y which is perpendicular to the rectangular side surfaces (with $1/|B_y|$ global decay).

Next, we interpret the clue of hinge state by comparing the assumption of side surfaces and the experimental data.

Firstly, when $|B_z|$ is applied, the bulk supercurrent will follow the Fraunhofer curve, i.e., the $1/|B_z|$ global decay. Therefore, a strong suppression of the bulk supercurrent can be realized at the side lobes (a manifestation of the partial cancellation of the positive and negative supercurrent), highlighting the boundary supercurrent. Of course, the larger the serial number $|n|$ of the side lobes is, the heavier the suppression of the bulk supercurrent. As a result, the experimental boundary- I_c could be extracted approximatively from the height of side lobes (with $1/|B_z|$ suppression of the bulk supercurrent) in Fig. 2a and S5 for device D1, at $B_y = 0 \text{ T}$ and $B_y = 0.2 \text{ T}$, respectively. The extracted boundary- I_c is shown in Fig. 5c.

Secondly, if there is a superconducting proximity effect on the side surfaces, we expect a Fraunhofer-like curve $I_c(B_y)/I_c(B_y = 0 \text{ T}) = |\sin(\pi\Phi/\Phi_0)/(\pi\Phi/\Phi_0)|$ in B_y perpendicular to the side surface. So, we can calculate the theoretical boundary- I_c at $B_y = 0.2 \text{ T}$, as shown by the pink triangles in Fig. 5c. However, experimental boundary- $I_c(B_y = 0.2 \text{ T})$ (blue squares) are much larger than the calculated values (pink triangles), as shown in Fig. 5c, which is in contrast with a side-surface supercurrent. Furthermore, boundary- $I_c(B_y = 0.2 \text{ T})$ are comparable to the boundary- $I_c(B_x = 0.2 \text{ T})$ (red circles), which does not support the side-surface scenario, either,

because a large suppression of the supercurrent caused by the Fraunhofer-like decay is expected for B_y , but not for B_x .

Theoretically, our detailed calculations show that NiTe₂ has the unconventional nature of charge mismatch, which gives rise to the obstructed hinge states. Importantly, due to the co-existence of time reversal symmetry and mirror symmetry, the electron spins of the hinge states are locked to be in the plane perpendicular to the hinges (along the x direction), which could explain the filter effect of the supercurrent under B_x . As illustrated in Fig. R2 shown above and Fig. 6h of the revised manuscript, this locking protects the Cooper pairs of the hinge states from undergoing depairing in B_x . In contrary, the spins of the bulk states are randomized without such protection, and hence the filtering of the hinge supercurrent can be observed. B_y represents the magnetic field perpendicular to the current. In this case, the acquired Zeeman energy to break Cooper pairs for the spins of the hinges is almost the same as that for the bulk. As a result, the Cooper pairs are not protected by the spin-momentum locking. Therefore, the magnetic field filtering of supercurrent is absent for B_y .

Therefore, combining the experimental and theoretical results, we reached a self-consistent conclusion that the supercurrent is carried by hinge states. On the other hand, the side-surface scenario contradicts with both the experimental data and the theoretical calculations, and we do not know how to explain the filtering effect merely based on the edge accumulation or other effects, like in graphene or Bi₂O₂Se systems.

4-2. What is the localization length of the hinge states on the side surfaces?

Reply: We thank Reviewer #4 for putting forward this question. The localization length of the hinge states on the side surface is important. However, to be honest, it is difficult to measure it merely from the transport measurement.

Fig. R4. **a**, Simulation of the SIP (with 9 side lobes shown) for the current density distribution displayed in the inset. **b**, Current density profiles extracted from **a**, assuming 9, 5, and 2 side lobes, respectively. The dashed lines are fittings using the Gaussian function.

In Josephson junctions, in principle, the localization length of the edge/hinge states could be extracted from the current density profiles obtained by the Dynes-Fulton approach. As many references have done, the length is 100 nm for the hinge states in WTe_2 [Nat. Mater. 19, 974-979, (2020)], 400 – 600 nm for the hinge states in Cr_3As_2 [Phys. Rev. Lett. 124, 156601 (2020)], 300 – 400 nm for the edge states in HgTe/CdTe [Nat. Phys. 10, 638-643, (2014)], and 220 nm for the edge states in graphene [Nat. Phys. 12, 128-133, (2016)]. Using the same method, we could estimate the localization length, which is around 150 nm for Fig. 2b, 190 nm for Fig. 3b, and 270 nm for Fig. 3d. These values are similar to those in the above references.

However, we should note that the resolution of this method is intimately related to the number of measured side lobes in the SIP, and thus the extracted length is not an accurate value, usually largely overestimated. One typical example can be seen from Figs. 3b and 3d, which present a large difference in the width of the current-density peaks due to the different number of side lobes. In order to describe the limitation of the Dynes-Fulton approach in extracting the localization length of the hinge/edge states, we show the simulation results in Fig. R4. We assume that the localization length of the hinge states is $d = 1\%W$ (W is the width of the junction), as shown by the inset of Fig.

R4a. The theoretical SIP containing SQUID-like component is shown in Fig. R4a. Then we take part of this SIP to extract the current density profile, as shown in Fig. R4b for 2, 5, and 9 side lobes on each side of the central lobe, respectively. The dashed lines are the Gaussian fitting of the peaks, by which the width can be obtained. However, the width is $5d$ for 9 side lobes, $9.5d$ for 5 side lobes, and $23d$ for 2 side lobes. Therefore, this value extracted from the Dynes-Fulton approach is an upper bound due to the limited number of side lobes in the SIP, and this approach is qualitative instead of quantitative in realistic experiments.

In fact, the localization length of hinge states should be small, on the scale of nanometers. For instance, the higher-order topological hinge states in Bi have been studied by STM and transport, e.g., in Nat. Phys. 14, 918-924 (2018), Nat. Commun. 8, 15941 (2017), and Science 364, 1255–1259 (2019). For the transport measurement, the hinge states can be recognized due to the different enclosed areas in a SQUID device for a narrow Bi nanowire with a thickness and width between 30 and 200 nm. So, the localization length should be much less than 30 nm, which is in fact ~ 1 nm obtained from the decay of the supercurrent. A more direct and accurate measurement is the STM, which presented the nanometer-scale hinge states clearly. Therefore, in Josephson junctions, the extracted 100 nm-wide hinge states in WTe₂ [Nat. Mater. 19, 974-979, (2020)], and the > 100 nm-wide hinge states in our work, should both be largely overestimated and both be in the nanometer scale.

Modification #16: In order to include a detailed discussion on the localization scale of the hinge states, we added the analysis shown above to the Supplementary Information Section VIII.

4-3. How can the authors distinguish the hinge states from edge states for such thin samples?

Reply: We sincerely thank Reviewer #4 for putting forward this essential question. The reviewer seems to worry about the extracted large localization length of the hinge states

(> 100 nm) from the Dynes-Fulton approach, which is even larger than the thickness of the sample (30 nm). As we have explained, the extracted length is largely overestimated, and it should be in the nanometer scale. One more example is that the localization length of the surface states of Bi₂Se₃ is also in the nanometer scale, since the top and bottom surfaces will be completely separated for a thickness of ~7 nm (7 quintuple layers) [Nat. Phys. 6, 584–588 (2010)]. In our work, the thickness of our samples is around 30 nm. When the in-plane magnetic field (B_y) is applied perpendicular to the current, the supercurrent carried by the hinge states and edge states (side surfaces) would exhibit a remarkably different critical field. The supercurrent on side surfaces should show much faster $1/|B_y|$ global Fraunhofer-like decay. However, the supercurrent on hinge states could persist to a higher magnetic field. It is one of the key points to distinguish the hinge states from edge states (side surfaces). For example, the supercurrent running through the hinge states of Bi nanowires (with a width of 30 nm – 200 nm) could sustain up to several Tesla [Nat. Phys. 14, 918-924 (2018)] (Ref. 13). In our work, as shown above in the reply to Comment 4-1, the experimental value of I_c at $B_y = 0.2$ T is much larger than the theoretical value, which is in contrary to the side-surface (edge states) scenario. The thickness of our plates of 30 nm also hints that the localization length of hinge states is much less than 30 nm, which is consistent with the expectation.

Overall, the experimental results are quite good. However, the theoretical explanation is not clear enough. Currently, there is not enough experimental nor theoretical evidence to support the presence of hinge states.

Reply: We sincerely appreciate Reviewer #4 for the positive comment about our experimental results. We are also grateful to the reviewer for the very constructive comments, which stimulated us to improve the quality and clarity of our paper a lot. We have made several significant modifications to express our results more explicitly and completely, and to reach a better presentation both experimentally and theoretically. These modifications include:

Modification #3: We interpret the clue of the hinge states by comparing the assumption of side surfaces and the experimental data in more detail. Our main explanation is that the experimental critical boundary supercurrent (boundary- I_c) at finite B_y is much larger than the calculated values if assuming a side-surface supercurrent, and hence it points to the hinge supercurrent.

Modification #7: We discuss in more detail the mechanism of the magnetic field filtering effect. We firmly believe that the key mechanism behind “magnetic field filtering” lies in the spin-momentum locking of the hinge states, as supported by our theoretical calculations and elaborated in paragraph 2, page 9. It is crucial to note that the magnetic field filtering of the supercurrent is effective only when the magnetic field is oriented along the Josephson current, denoted as B_x .

Modification #10: We attribute the small in-plane magnetic field required for suppressing the bulk supercurrent to the Gaussian-like decay of the bulk supercurrent, which is caused by the spin-flip scattering when the magnetic field is parallel to the current [Phys. Rev. B 76, 064514 (2007)]. A detailed explanation was added.

Finally, we would like to introduce our train of thought briefly again, as shown in Fig. R3. Experimentally, we observed the transition from the bulk to the edge, and found contrary with the side-surface scenario. Therefore, it points to the hinge supercurrent. Theoretically, our calculations show the obstructed hinge states with spin-momentum locking, which explains the observed filtering effect well. Therefore, we reached a self-consistent conclusion that the supercurrent is carried by hinge states.

We are very grateful to all four reviewers for the very constructive and valuable comments and suggestions, which helped us a lot to notice the incompleteness, missed interpretations and ambiguities, and stimulated us to make many modifications to improve the manuscript. We appreciate this opportunity to consider the underlying physics and the presentation more thoroughly. With all these modifications and responses, we hope that we have answered all the

questions properly, and have clarified our experimental and theoretical results on the hinge states and the filtering effect.

REVIEWER COMMENTS

Reviewer #1 (Remarks to the Author):

I would like to thank the authors for their answers to my questions and concerns. I still have a concern regarding the $1/|B|$ decay analysis. I agree for the ideal Fraunhofer pattern, the magnitude of the lobes will follow the $1/B$ decay. However, there are lots of examples of SIPs that don't follow $1/B$ decay even without edge supercurrents, for example (Nano Lett 2017, 17, 10, 6125–6130, Phys. Rev. B 107 094522 (2023), Nat. Nanotechnol. 17, 39-44 (2022)). Moreover, the nonuniformity of supercurrent distribution can also cause the magnitude of the lobes deviating from the ideal $1/B$ decay as mentioned in ref. 11 in the manuscript. And such nonuniformity is inevitable due to for example imperfect fabrication process, as well as the existence of long Josephson junctions, etc., therefore, most Josephson junctions won't exactly follow the $1/B$ decay.

The authors mentioned in rebuttal letter "...if there is a superconducting proximity effect on the side surfaces, we expect a Fraunhofer-like curve $I_c(B_y)/I_c(B_y = 0 \text{ T}) = |\sin(\pi\Phi/\Phi_0)/(\pi\Phi/\Phi_0)|$ in B_y perpendicular to the side surface..." However, there's no theory that supports this claim. While applying B_y , the edge states of the top and bottom surfaces, if exist, can also cause the SIP deviate from ideal $1/B$ decay, as well as other extrinsic factors that mentioned above.

The edge and hinge currents are notoriously difficult to distinguish, as described in Nat. Mater. 19, 974–979 (2020), ref. 15 in the manuscript. They did a lot more effort to distinguish them, the simple $1/B$ decay analysis is far from plausible to distinguish.

The $1/B$ decay analysis will ruin the novelty and rigorousness of the manuscript. I would be happy to recommend publishing at Nature Communications if the authors can delete the $1/B$ analysis and weaken their hinge supercurrent claim, or they can provide stronger evidence as described in ref. 15.

Reviewer #2 (Remarks to the Author):

The revised manuscript has shown significant improvement, and I appreciate the authors' thorough responses to my previous comments. However, I have one more query related to crystal anisotropy. In the study, the authors examine bulk supercurrent behavior using the geometry depicted in the inset of Figure 5a. I noticed that the current direction in this geometry differs from the D1 direction as illustrated in Figure 2. This raises concerns about the crystal anisotropy of NiTe₂.

I would like to inquire whether there are any relevant experimental findings regarding the anisotropic or isotropic transport behavior in NiTe₂. Additionally, I propose that the authors consider employing the geometry introduced in Figure 3a of the publication [National Science Review 7, 1468-1475, 7,9, (2020)] to address and clarify the issue of anisotropy. This could potentially enhance the comprehensibility of the research findings.

Reviewer #3 (Remarks to the Author):

Following this first round of review and rebuttal, I find that one problem still remains to be addressed before this work can be accepted. As stated by myself and reviewer 4, the precise mechanism for bulk suppression should be discussed. Relating to this issue, I find several points which the authors should address.

There appears to be a difference between the suppression rates of $I_{c_bulk}(B_x)$ and $I_{c_bulk}(B_y)$. In fact, this point was raised specifically by reviewer 4, and has not been understood by the authors. They have provided a lengthy explanation of the hinge mode being immune to B_x – and not to B_y – but this is really not the issue.

Well, as the authors argue, they can distinguish bulk and edge contributions. So a possible way to address this confusion would be to separate the hinge and bulk contributions (based on the Dynes-Fulton approach) to show $I_{c_bulk}(B_x)$, $I_{c_bulk}(B_y)$, and I_{c_hinge} for both.

I believe the BULK responds differently to orientation. The question is therefore, given that the bulk states are not immune to the in-plane field, their response appears to be anisotropic. As seen in Figure 4, $B_x = 0.04$ T gives rise to $I_c = 2 \mu A$. However, $B_y = 0.04$ T gives rise to $I_c = 7-8 \mu A$.

I am not fully satisfied by the explanation given by the authors for the bulk suppression. They cite PRB 76 064514. According to this paper, there should be a suppression of critical current due to magnetically induced spin-flips. However, I am not convinced that this model is the one which dictates $I_c(B)$ in the present case. Specifically, as written in the PRB, the B^2 dependence follows from the dependence $D e^2 B^2 W^2 / 6 \hbar$. This is exactly the expression for depairing in a thin film in a diffusive system. (W – thickness, D – diffusion coefficient. see e.g. the book by Tinkham, chapter 10). It originates from the screening currents and depends on the sample geometry. I doubt that this is the mechanism at play here. W (thickness) remains the same for B_x and for B_y in this “Orbital spin-flip” mechanism.

An alternative mechanism would be found due to finite momentum of the order parameter. This “FFLO”-like mechanism was discussed in Phys. Rev. B 103, 115401. This “Zeeman-driven FFLO” mechanism should be symmetric towards B_x and B_y .

A third mechanism, as discussed in the Nature Physics paper of S. Hart (Yacoby group) is related to Spin-Orbit suppression. Here, one has to assume the presence of a SO term and model the suppression based on the type of SO expected. I do not ask of the authors to reproduce the complex arguments presented in Hart et al., (and a related paper by the N. Mason group) – but I believe they should address this possibility.

Reviewer #4 (Remarks to the Author):

As mentioned in the previous report, the experimental findings of the authors are very interesting. The authors clearly demonstrated the presence of supercurrent along the edges of NiTe₂. The most intriguing finding is that the bulk supercurrent can be readily suppressed by an in-plane magnetic

field applied parallel to the direction of the supercurrent.

In the earlier version of the manuscript, the authors did not provide a satisfactory explanation as to why the bulk supercurrent could be easily suppressed by an in-plane magnetic field, while the edge supercurrents are protected from the Zeeman field (oriented along the x-direction). In the revised version, the authors proposed that spin-orbit coupling (SOC), analogous to the Ising SOC present in transition metal dichalcogenides, is responsible for protecting the edge supercurrents. I found this explanation to be reasonable. [Concerning Ising superconductivity, the following work should be cited along with Ref. 49 and Ref. 50: Nature Physics 12, 139-143 (2016).]

On the theoretical side, the authors could have performed simple model calculations demonstrating how the Zeeman field suppresses the bulk supercurrent and how SOC protects the edge supercurrent. However, given that this is primarily an experimental work, the current level of explanation and reasoning is likely acceptable to readers.

Overall, I found this work to be very interesting and I support its publication in Nature Communications.

We appreciate the valuable insights provided by all the reviewers. These important concerns and suggestions helped us a lot to further improve our manuscript. We are grateful to see that Reviewer #4 has considered that this is primarily an experimental work, the current level of explanation and reasoning is likely acceptable to readers. We sincerely thank Reviewer #4 for supporting its publication in Nature Communications.

In the revised version of our manuscript, we have tried to address all the concerns raised by all the reviewers. We deleted the $1/|B|$ decay analysis from the main text and weakened the tone of the hinge supercurrent claim, as suggested by Reviewer #1. We also performed control experiment and added the results regarding the influence of crystal anisotropy of NiTe₂, as proposed by Reviewer #2. Furthermore, we added a more complete and inclusive explanation of the anisotropic suppression rate and ultra-low critical field along B_x , as commented by Reviewer #3 and Reviewer #4. The followings are our point-by-point responses to the reviewers' reports. The words in **black** font color are the reviewers' reports, in **blue** font are our responses. The modifications in the manuscript and the Supplementary Information are indicated in **red**.

Reviewer #1

I would like to thank the authors for their answers to my questions and concerns. I still have a concern regarding the $1/|B|$ decay analysis. I agree for the ideal Fraunhofer pattern, the magnitude of the lobes will follow the $1/B$ decay. However, there are lots of examples of SIPs that don't follow $1/B$ decay even without edge supercurrents, for example (Nano Lett 2017, 17, 10, 6125–6130, Phys. Rev. B 107 094522 (2023), Nat. Nanotechnol. 17, 39-44 (2022)). Moreover, the nonuniformity of supercurrent distribution can also cause the magnitude of the lobes deviating from the ideal $1/B$ decay as mentioned in ref. 11 in the manuscript. And such nonuniformity is inevitable due to for example imperfect fabrication process, as well as the existence of long

Josephson junctions, etc., therefore, most Josephson junctions won't exactly follow the $1/B$ decay.

The authors mentioned in rebuttal letter "...if there is a superconducting proximity effect on the side surfaces, we expect a Fraunhofer-like curve $I_c(B_y)/I_c(B_y = 0 \text{ T}) = |\sin(\pi\Phi/\Phi_0)/(\pi\Phi/\Phi_0)|$ in B_y perpendicular to the side surface..." However, there's no theory that supports this claim. While applying B_y , the edge states of the top and bottom surfaces, if exist, can also cause the SIP deviate from ideal $1/B$ decay, as well as other extrinsic factors that mentioned above.

Reply: We would like to express our sincere gratitude to Reviewer #1 for providing insightful comments regarding the $1/|B|$ decay analysis. We fully concur that in certain Josephson junctions exhibiting non-ideal Fraunhofer patterns, the magnitude of the lobes will not exactly follow the $1/|B|$ decay. The three references provided by the reviewer involve the self-field effect when the critical current is very large [Nano Lett 2017, 17, 10, 6125–6130], the irregular shape of sandwich-like junctions [Phys. Rev. B 107 094522 (2023)], and an asymmetry in the current-phase relation [Nat. Nanotechnol. 17, 39-44 (2022)]. These examples may not be directly applicable to our rectangular side-surface-based junctions, though, they do demonstrate different scenarios of the non-Fraunhofer patterns. Of course, there are also planar Josephson junctions with a regular rectangular shape that exhibit an approximate $1/|B|$ decay, such as Sci. Rep. 2, 1-5 (2012), Nat. Phys. 10, 638-643 (2014), Nat. Phys. 13, 87-93 (2017), etc. Furthermore, we also agree that the nonuniform supercurrent distribution can cause deviations in the magnitude of the lobes from the ideal $1/|B|$ decay. However, it is usually hard to tell the detailed nonuniformity of a real device due to for example imperfect fabrication process. Indeed, we cannot exclude the potential for non-ideal Fraunhofer patterns in our devices, stemming from factors like imperfect fabrication process, long Josephson junctions, and different current channels. We agree with the reviewer's comments. To be more rigorous, as suggested by the reviewer, we have deleted the $1/|B|$ decay analysis from the main text.

The edge and hinge currents are notoriously difficult to distinguish, as described in Nat. Mater. 19, 974–979 (2020), ref. 15 in the manuscript. They did a lot more effort to distinguish them, the simple $1/B$ decay analysis is far from plausible to distinguish.

The $1/B$ decay analysis will ruin the novelty and rigorousness of the manuscript. I would be happy to recommend publishing at Nature Communications if the authors can delete the $1/B$ analysis and weaken their hinge supercurrent claim, or they can provide stronger evidence as described in ref. 15.

Reply: We appreciate Reviewer #1 for these valuable comments. The work, Nat. Mater. 19, 974–979 (2020), is very nice and proposes an ingenious method to distinguish the edge and hinge currents. We agree that the $1/|B|$ decay analysis may not be a reliable means of differentiation. There could be various factors that induce deviations from the ideal $1/|B|$ decay, as suggested by the reviewer. Following the reviewer’s suggestions, we have deleted the $1/|B|$ decay analysis including Fig. 5c from the main text in the revised version.

In the last round review, we also modified the paragraph in the Supplementary Information that explains such analysis. After a careful consideration based on the reviewer’s comments, we think that it would be beneficial for the readers to see the analysis of the data when B_y is applied, incorporating the various mechanisms, as mentioned by the reviewer, that induce deviations from the ideal Fraunhofer patterns. Therefore, we moved Fig. 5c to the Supplementary Information, and added the discussion on the different mechanisms. We hope that the reviewer will agree that such discussion is helpful for the readers.

Regarding the hinge supercurrent claim, we agree with Reviewer #1 that the statement could be weakened. Following the reviewer’s suggestion, to enhance the rigor of this work, we have weakened the tone of the hinge supercurrent and presented a more tempered claim, by saying boundary instead of hinge for some places in the manuscript and adding the word “probable” in the abstract and the conclusion part. We sincerely

hope that the reviewer could be satisfactory with these modifications.

Modification #1: We deleted the $1/|B|$ decay analysis from the main text, moved Fig. 5c from the main text to the Supplementary Section IV, and added the discussion on the different mechanisms that induce deviations from the ideal $1/|B|$ decay.

Modification #2: We weakened the tone of the hinge supercurrent and presented a more tempered claim, by saying boundary instead of hinge for some places in the manuscript and adding the word “probable” in the abstract and the conclusion part.

Reviewer #2:

The revised manuscript has shown significant improvement, and I appreciate the authors' thorough responses to my previous comments. However, I have one more query related to crystal anisotropy. In the study, the authors examine bulk supercurrent behavior using the geometry depicted in the inset of Figure 5a. I noticed that the current direction in this geometry differs from the D1 direction as illustrated in Figure 2. This raises concerns about the crystal anisotropy of NiTe₂.

I would like to inquire whether there are any relevant experimental findings regarding the anisotropic or isotropic transport behavior in NiTe₂. Additionally, I propose that the authors consider employing the geometry introduced in Figure 3a of the publication [National Science Review 7, 1468-1475, 7,9, (2020)] to address and clarify the issue of anisotropy. This could potentially enhance the comprehensibility of the research findings.

Reply: We appreciate Reviewer #2 for raising the issue of crystal anisotropy in NiTe₂. Regrettably, we did not find any references addressing the anisotropic or isotropic transport behavior in NiTe₂. We regret any oversight in this regard. Our primary objective in this study is to elucidate the contribution of the boundary supercurrent in

our Josephson junctions (JJs), as well as the magnetic filtering. It is essential to clarify the role of the sample hinges/side surfaces. Therefore, we fabricated a JJ whose junction region did not include the hinges/side surfaces of the sample, as shown by the left inset of Fig. 5a. In this case, no boundary supercurrent could be seen, supporting the observation of boundary supercurrent in other devices.

We are grateful to Reviewer #2 for pointing out that “the current direction in this geometry differs from the D1 direction as illustrated in Figure 2”. We diligently followed the reviewer’s suggestion and fabricated the device with the current direction the same as the other devices. The results are shown in Fig. R1. Please note that the exfoliated NiTe₂ nanoplates usually have parallel long edges, indicating a certain crystal direction. The junctions (D1, D2-1, D3-1, D4-1, and D5) that cover the sample hinges/side surfaces and present boundary supercurrent are vertical to the parallel long edges of the sample, and the junction (D2-2) that does not cover the hinges/side surfaces and does not show boundary supercurrent is parallel to the parallel long edges. Therefore, to answer the reviewer’s question, we fabricated a junction (D6) that is vertical to the parallel long edges of the nanoplate, as shown in the inset of Fig. R1a. We observed similar patterns to junction D2-2 (Fig. 5), showing neither boundary supercurrent nor anisotropic behavior. Please note that the current density is not uniform as shown by the right inset of Fig. 5a, and thus the SIP is distorted from the standard Fraunhofer pattern and side lobes can be hardly observed, as discussed in the last round review.

Fig. R1 a, SIP for D6 at 10 mK without in-plane magnetic field. **b**, SIP for D6 at 10 mK under $B_x = 0.026$ T.

Modification #3: We added Fig. R1 and related discussion to the Supplementary Section VIII.

Reviewer #3:

Following this first round of review and rebuttal, I find that one problem still remains to be addressed before this work can be accepted. As stated by myself and reviewer 4, the precise mechanism for bulk suppression should be discussed. Relating to this issue, I find several points which the authors should address.

There appears to be a difference between the suppression rates of $I_{c_bulk}(B_x)$ and $I_{c_bulk}(B_y)$. In fact, this point was raised specifically by reviewer 4, and has not been understood by the authors. They have provided a lengthy explanation of the hinge mode being immune to B_x – and not to B_y – but this is really not the issue. Well, as the authors argue, they can distinguish bulk and edge contributions. So a possible way to address this confusion would be to separate the hinge and bulk contributions (based on the Dynes-Fulton approach) to show $I_{bulk}(B_x)$, $I_{bulk}(B_y)$, and I_{hinge} for both. I believe the bulk responds differently to orientation. The question is therefore, given that the bulk states are not immune to the in-plane field, their response appears to be anisotropic. As seen in Figure 4, $B_x = 0.04$ T gives rise to $I_c = 2\mu A$. However, $B_y = 0.04$ T gives rise to $I_c = 7-8 \mu A$. I am not fully satisfied by the explanation given by the authors for the bulk suppression. They cite PRB 76 064514. According to this paper, there should be a suppression of critical current due to magnetically induced spin-flips. However, I am not convinced that this model is the one which dictates $I_c(B)$ in the present case. Specifically, as written in the PRB, the B^2 dependence follows from the dependence $D e^2 B^2 W^2 / 6 \hbar$. This is exactly the expression for depairing in a thin film in a diffusive system. (W – thickness, D – diffusion coefficient. see e.g. the book by Tinkham, chapter 10). It originates from the screening currents and depends on the sample

geometry. I doubt that this is the mechanism at play here. W (thickness) remains the same for B_x and for B_y in this “Orbital spin-flip” mechanism.

An alternative mechanism would be found due to finite momentum of the order parameter. This “FFLO”-like mechanism was discussed in Phys. Rev. B 103, 115401. This “Zeeman-driven FFLO” mechanism should be symmetric towards B_x and B_y .

A third mechanism, as discussed in the Nature Physics paper of S. Hart (Yacoby group) is related to Spin-Orbit suppression. Here, one has to assume the presence of a SO term and model the suppression based on the type of SO expected. I do not ask of the authors to reproduce the complex arguments presented in Hart et al., (and a related paper by the N. Mason group) – but I believe they should address this possibility.

Reply: We extend our sincere appreciation to Reviewer #3 for his/her invaluable comments and suggestions, which have significantly enhanced our understanding of the mechanism regarding to bulk-supercurrent suppression. We agree with that the suppression rates of $I_c^{\text{Bulk}}(B_x)$ and $I_c^{\text{Bulk}}(B_y)$ differ, as shown in Fig. R2 the supercurrent density profiles of Fig. 4b and Fig. 4c, and we realize that this distinction was not explicitly addressed in the previous manuscript. In the last revision, we only considered the “orbital spin-flip” mechanism to account for the ultra-low critical field along B_x , a concept also mentioned in Nat. Commun. 11, 1150 (2020). However, the reviewer's feedback has prompted us to realize that the current mechanism is insufficient to account for the anisotropic bulk suppression rate, especially when considering the same width (W) for both B_x and B_y .

Indeed, as the reviewer pointed out, the data shows an anisotropic suppression on the bulk supercurrent. As seen in Fig. 4, $B_x = 0.04$ T gives rise to $I_c = 1.6$ μA . However, $B_y = 0.04$ T gives rise to $I_c = 7$ μA . The anisotropy can also be found by the Dynes-Fulton approach, as suggested by the reviewer. As shown in Fig. R2, boundary supercurrent

Fig. R2 Supercurrent density profiles $J_s(y)$ of D4-1 at $B_x = 0.04$ T and $B_y = 0.04$ T, respectively. The bulk supercurrent at $B_x = 0.04$ T is suppressed more largely than at $B_y = 0.04$ T.

dominates at $B_x = 0.04$ T, while the bulk supercurrent dominates at $B_y = 0.04$ T. We noticed that for the “orbital spin-flip” mechanism, the B^2 dependence not only relies on the thickness W , but is also associated with the diffusion coefficient D . Therefore, while W is the same for B_x and B_y , the anisotropic suppression of the bulk supercurrent requires D to be anisotropic in NiTe₂, if this mechanism indeed applies. As of now, no reports on the anisotropic diffusion coefficient in NiTe₂ have been published, indicating that the “orbital spin-flip” mechanism may not capture the anisotropic suppression, as suggested by the reviewer.

On the other hand, we agree with that the “orbital spin-flip” is not the only possible mechanism to explain the anisotropic bulk suppression rates. We appreciate the reviewer for suggesting other alternative explanations.

As reported in Phys. Rev. B 103, 115401 (2021), the application of an in-plane magnetic field can lead to Zeeman splitting, resulting in an exponentially suppressed critical current, aligning with the observed fast bulk suppression rate. Moreover, this Zeeman-driven mechanism is also intricately linked to the diffusion coefficient (D). As the reviewer mentioned, this “Zeeman-driven FFLO” mechanism should be symmetric towards B_x and B_y . If this mechanism applies to the anisotropic suppression rates

between B_x and B_y in our work, again, anisotropic D in NiTe₂ is required, which is unknown yet.

In addition, we would like to express our gratitude to the reviewer for suggesting another mechanism related to spin-orbital suppression. In the nice work done by Hart et al. [Nat. Phys. 13, 87, (2017)], finite-momentum Cooper pairing was observed in HgTe quantum wells in an in-plane magnetic field. As the reviewer mentioned, spin-orbit coupling (SOC) resulted from structural inversion asymmetry or bulk inversion asymmetry is required, and the performance of the Josephson junction depends on the type of SOC. Detailed calculations involving SOC and the geometry of the junction did show a fast and anisotropic suppression of the bulk supercurrent in an in-plane magnetic field. The spin-orbital suppression is a manifestation of the interplay between in-plane magnetic field and SOC, and the anisotropy mainly comes from the difference between B_x and B_y on the flux penetration. For the scenario when B_x aligns parallel to the current direction, it allows the magnetic flux to penetrate the superconducting electrodes and subsequently suppresses the supercurrent. This mechanism could indeed effectively account for both our observed anisotropic in-plane suppression behavior and the remarkably low critical in-plane field along B_x , while the type of SOC requires further investigation. We thank the reviewer again for providing this very constructive suggestion which is mostly likely the physical mechanism. In addition, we would like to note that the observed boundary supercurrent in both zero and finite in-plane magnetic fields is particular in our work.

Modification #4: In order to address all the possibilities for the anisotropic suppression rate and the ultra-low critical field along B_x , we have extended the discussion on these different mechanisms in Supplementary Section III.

Reviewer #4:

As mentioned in the previous report, the experimental findings of the authors are very interesting. The authors clearly demonstrated the presence of supercurrent along the edges of NiTe₂. The most intriguing finding is that the bulk supercurrent can be readily suppressed by an in-plane magnetic field applied parallel to the direction of the supercurrent.

In the earlier version of the manuscript, the authors did not provide a satisfactory explanation as to why the bulk supercurrent could be easily suppressed by an in-plane magnetic field, while the edge supercurrents are protected from the Zeeman field (oriented along the x-direction). In the revised version, the authors proposed that spin-orbit coupling (SOC), analogous to the Ising SOC present in transition metal dichalcogenides, is responsible for protecting the edge supercurrents. I found this explanation to be reasonable. [Concerning Ising superconductivity, the following work should be cited along with Ref. 49 and Ref. 50: Nature Physics 12, 139-143 (2016).]

On the theoretical side, the authors could have performed simple model calculations demonstrating how the Zeeman field suppresses the bulk supercurrent and how SOC protects the edge supercurrent. However, given that this is primarily an experimental work, the current level of explanation and reasoning is likely acceptable to readers.

Overall, I found this work to be very interesting and I support its publication in Nature Communications.

Reply: We sincerely thank Reviewer #4 for supporting the publication in Nature Communications. Regarding the suppression of the supercurrent in an in-plane magnetic field, especially on the anisotropy, Reviewer #3 constructively suggested other possible mechanisms. We added the discussion on these different scenarios in Supplementary Section III. We also thank the reviewer for suggesting a new reference. The paper Nature Physics 12, 139-143 (2016) has been cited in the revised manuscript.

Modification #5: We added Nature Physics 12, 139-143 (2016) as Ref. 51.

REVIEWERS' COMMENTS

Reviewer #1 (Remarks to the Author):

I believe my concerns have been addressed by the authors and I am happy to recommend publishing on Nature Communications with current version. Since the authors have weakened their claims of hinge state, if the authors could also weaken the hinge state in their title, by adding word 'probable' or change to 'boundary state', that would be the best!

Reviewer #2 (Remarks to the Author):

I would like to thank the authors for their efforts and answers to my question. The authors' works to find out anisotropy of NiTe₂ resolves my concern and I am happy to recommend the revised manuscript for publication in Nature Communications.

Reviewer #3 (Remarks to the Author):

I believe the manuscript now could be scientifically correct, but the overall level of presentation became quite patchy, as response to referees resulted in removal of large segments of the text without the introduction of a replacement scientific content.

The section responding to the suppression mechanism of the bulk has largely been delegated to the supplementary. I think this is a mistake.

We appreciate the constructive insights provided by all the reviewers. The opportunity to further enhance our paper is extremely valuable. We are grateful to see that Reviewer #1 and #2 have recommended its publication in Nature Communications.

The followings are our point-by-point responses to the reviewers' reports. The words in **black** font color are the reviewers' reports, in **blue** font are our responses. The modifications in the manuscript are indicated in **red**.

Reviewer #1:

I believe my concerns have been addressed by the authors and I am happy to recommend publishing on Nature Communications with current version. Since the authors have weakened their claims of hinge state, if the authors could also weaken the hinge state in their title, by adding word 'probable' or change to 'boundary state', that would be the best!

Reply: We would like to express our sincere gratitude to Reviewer #1 for recommending its publication. We also thank the reviewer for suggesting the modification of the title. To be more rigorous, we changed the word "hinge" to "boundary" in the title.

Reviewer #2:

I would like to thank the authors for their efforts and answers to my question. The authors' works to find out anisotropy of NiTe₂ resolves my concern and I am happy to recommend the revised manuscript for publication in Nature Communications.

Reply: We appreciate Reviewer #2 for recommending the publication of our paper.

Reviewer #3:

I believe the manuscript now could be scientifically correct, but the overall level of presentation became quite patchy, as response to referees resulted in removal of large segments of the text without the introduction of a replacement scientific content.

Reply: We would like to express our gratitude to Reviewer #3 for pointing out the patchiness of the presentation after the last-round revision. Indeed, following the suggestions of the reviewers we deleted the $1/|B|$ decay analysis from the main text, and extended the discussion in the supplementary. We agree that the flow of the presentation in the main text could be further improved. Therefore, in the revised version we added a short paragraph on page 7 to include such discussion.

Modification #1: On page 7 we added the following paragraph to reach a better presentation. “A clue of the boundary supercurrent may be found by comparing the assumption of side surfaces and the experimental data (Supplementary Section IV). The measured critical boundary supercurrent at finite B_y is much larger than the calculated values if assuming a side-surface supercurrent that follows the ideal Fraunhofer pattern. Diverse mechanisms could induce deviations, though, it indicates the possibility of hinge supercurrent.”

The section responding to the suppression mechanism of the bulk has largely been delegated to the supplementary. I think this is a mistake.

Reply: We appreciate Reviewer #3 for pointing out this issue. We agree that the explanation of the suppression mechanism of the bulk supercurrent is important, and deserves a section in the main text. However, we note that a detailed discussion is quite lengthy as has been shown in Supplementary Section III. Therefore, we believe that a concise version of the discussion in the main text would be appropriate, while keeping the detailed discussion in the supplementary.

Modification #2: On page 9 we added a paragraph which is a shortened version of Supplementary Section III to discuss the mechanisms for the suppression of the bulk supercurrent. “Next, we discuss the possible mechanisms for the fast suppression of the bulk supercurrent and the anisotropic rate for B_x and B_y (see a more detailed discussion in Supplementary Section III). 1) The fast suppression could be contributed to the Gaussian-like decay of the bulk supercurrent due to orbital spin-flip⁵⁴. However, the anisotropic rate requires an anisotropic diffusion constant (D) in NiTe₂, which is unknown. 2) The in-plane magnetic field can lead to Zeeman splitting, resulting in an exponentially suppressed critical current⁵². Again, an anisotropic D is required. 3) A likely mechanism is related to spin-orbital suppression, a manifestation of the interplay between in-plane magnetic field and spin-orbital coupling, and the anisotropy mainly comes from the difference between B_x and B_y on the flux penetration⁴³. This mechanism could indeed effectively account for both our observed anisotropic in-plane suppression behavior and the remarkably low critical in-plane field along B_x . 4) It might also come from the vimineous shape of the electrodes that exhibit anisotropic demagnetization⁵⁵⁻⁵⁷. Note that this mechanism was usually applied to intrinsic superconductors, while in our case it is proximity-induced JJ.”